# Sustainable Online Reinforcement Learning for Auto-bidding

**Zhiyu Mou**[1,2*], **Yusen Huo**[1], **Rongquan Bai**[1], **Mingzhou Xie**[1], **Chuan Yu**[1],
**Jian Xu**[1], **Bo Zheng**[1†]

[1] Alibaba Group, Beijing, China
[2] Department of Automation, Tsinghua University, Beijing, China
mouzy20@mails.tsinghua.edu.cn
{huoyusen.huoyusen, rongquan.br, mingzhou.xmz,
yuchuan.yc, xiyu.xj, bozheng}@alibaba-inc.com

## Abstract

Recently, auto-bidding technique has become an essential tool to increase the revenue of advertisers. Facing the complex and ever-changing bidding environments in the real-world advertising system (RAS), state-of-the-art auto-bidding policies usually leverage reinforcement learning (RL) algorithms to generate real-time bids on behalf of the advertisers. Due to safety concerns, it was believed that the RL training process can only be carried out in an offline virtual advertising system (VAS) that is built based on the historical data generated in the RAS. In this paper, we argue that there exists significant gaps between the VAS and RAS, making the RL training process suffer from the problem of *inconsistency between online and offline* (IBOO). Firstly, we formally define the IBOO and systematically analyze its causes and influences. Then, to avoid the IBOO, we propose a sustainable online RL (SORL) framework that trains the auto-bidding policy by directly interacting with the RAS, instead of learning in the VAS. Specifically, based on our proof of the Lipschitz smooth property of the Q function, we design a safe and efficient online exploration (SER) policy for continuously collecting data from the RAS. Meanwhile, we derive the theoretical lower bound on the safety degree of the SER policy. We also develop a variance-suppressed conservative Q-learning (V-CQL) method to effectively and stably learn the auto-bidding policy with the collected data. Finally, extensive simulated and real-world experiments validate the superiority of our approach over the state-of-the-art auto-bidding algorithm.

## 1   Introduction

In the era of Internet, online advertising business has become one of the main profit models for many companies, such as Google [6] and Alibaba [3], which, at the same time, benefits millions of advertisers who are willing to bid for impression opportunities. Contemporary online advertising systems, as auctioneers, usually have large amount of candidate advertisers contesting for numerous impression opportunities at every moment. Making auction decisions based on accurate evaluation of each impression opportunity for all advertisers within several milli-seconds is computationally infeasible. Therefore, a real-world advertising system (RAS) adopts a cascade architecture [7, 10]. In this architecture, the auction of each impression opportunity is completed through several stages. Without loss of generality, we here simply view the RAS as a system with two stages: stage 1 and stage 2. The auction process of each impression opportunity is as follows: in stage 1, rough

---

*Work was done during an internship at Alibaba Group.
†Corresponding author.

evaluations are conducted for all the candidate advertisers ($\sim 10^6$), and a group of the most promising advertisers ($\sim 10^2$) are fed to the stage 2; and in stage 2, accurate valuations and auction are carried out to determine the winning advertiser of the impression opportunity. The winner will gain the value of the impression opportunity and pay the market price. See Appendix A.1.1 for detailed explanations on the RAS structures. Faced with huge amount of impression opportunities at every moment, advertisers cannot bid in the granularity of individual opportunities. Recently, many auto-bidding policies have emerged to realize automatic real-time biddings for each impression opportunity on behalf of advertisers [1, 2, 4, 5], which significantly increase their revenues. State-of-the-art auto-bidding policies are usually learned with reinforcement learning (RL) algorithms [5]. It was believed that the auto-bidding policy being trained cannot directly access to the RAS during the RL training process due to safety concerns [1]. A common solution in most existing works [1, 2, 4, 5] is to train the auto-bidding policy in a virtual advertising system (VAS) — an offline simulated environment that is built based on the advertisers' historical data generated in the RAS. See Appendix A.1.2 for the details of the VAS structures.

**IBOO Challenges.** However, we argue that there exists gaps between the RAS and the VAS, which makes this common solution suffer from the problem of *inconsistencies between online and offline* (IBOO). Here, *online* refers to the RAS, while *offline* refers to the VAS, and *inconsistencies* refer to the gaps. Formally, we can define the IBOO as follows.

**Definition 1** (Inconsistencies Between Online and Offline, IBOO). *The IBOO refers to the gaps between the RAS and the VAS that prevent the VAS from accurately simulating the RAS.*

Specifically, there are two dominated gaps between the RAS and the VAS. One is that the VAS cannot accurately simulate the cascade auction architecture of the RAS. For example, due to the constraint on computing power, the VAS is built only based on the historical data generated in stage 2 of the RAS[3], which makes the VAS lack the mechanism of stage 1 in the RAS. In addition, the VAS does not incorporate the exact influences of other advertisers as RAS does, which makes up the second dominated gap. For example, in the RAS, the market prices are determined by the bids of all advertisers which can change during the training process. However, the VAS always provides constant market prices for the auto-bidding policy being trained. See figures in Appendix A.3 for illustrations. Essentially, the IBOO makes the VAS provide the auto-bidding policy with false rewards and state transitions during RL training process. As presented in Table. 1, the auto-bidding policy with better performance in the VAS (i.e., higher $R/R*$) may yield poorer performance in the RAS (i.e., lower A/B test value). Hence, the IBOO can seriously degrade the performance of the auto-bidding policy in the RAS. One way to address the IBOO challenge is to improve the VAS and reduce the gaps as much as possible. However, due to the complex and ever-changing nature of the RAS, the reduction in IBOO is usually very limited, resulting in little improvement in the auto-bidding policy performance. Hence, new schemes need to be devised so as to avoid the IBOO and improve the performance of auto-bidding policies. Besides, it is worth noting that the IBOO resembles the *sim2real* problem studied in other realms, such as computer visions [8] and robotics [9]. Nonetheless, to the best of our knowledge, this is the first work that formally put forward the concept of IBOO in the field of auto-bidding and systematically analyzes and resolves it.

Table 1: Influence of IBOO. $R/R*$ and A/B test values are the performance evaluations of the auto-bidding policies in the VAS and RAS, respectively. Both are the higher the better. We rank the ten policies accordingly. $\uparrow$ means that the performance in the RAS is higher than that in the VAS, while $\downarrow$ means the opposite. $-$ means that the evaluations in the RAS and VAS are the same.

| Policy | $R/R*$ (rank) | A/B Tests (rank) | Policy | $R/R*$ (rank) | A/B Tests (rank) |
|--------|---------------|------------------|--------|---------------|------------------|
| 1 | 0.9118 (1) | $-2.50\%$ (9) $\downarrow$ | 6 | 0.8563 (6) | $+4.30\%$ (2) $\uparrow$ |
| 2 | 0.8731 (2) | $+3.10\%$ (6) $\downarrow$ | 7 | 0.8535 (7) | $+4.40\%$ (1) $\uparrow$ |
| 3 | 0.8656 (3) | $-3.20\%$ (10) $\downarrow$ | 8 | 0.8434 (8) | $-1.50\%$ (8) $-$ |
| 4 | 0.8656 (4) | $+1.20\%$ (7) $\downarrow$ | 9 | 0.8428 (9) | $+3.60\%$ (5) $\uparrow$ |
| 5 | 0.8594 (5) | $+4.30\%$ (2) $\uparrow$ | 10 | 0.8111 (10) | $+3.80\%$ (4) $\uparrow$ |

In this paper, we propose a novel sustainable online reinforcement learning (SORL) framework to address the IBOO challenge. For the first time, the SORL abandons the way of learning with the VAS and trains the auto-bidding policy by directly interacting with the RAS. Notably, the SORL can obtain true rewards and state transitions from the RAS and thereby does not suffer from the IBOO problem.

---

[3]Detailed reasons are shown in Appendix A.1.2.

The SORL contains two main algorithms. The first one is a *safe and efficient* online exploration policy for collecting data from the RAS, named as the SER policy. Specifically, to guarantee the safety of explorations, we design a safety zone to sample actions from based on the Lipschitz smooth property of the Q function we theoretically proved. We also derive the lower bound on the safety degree. To increase the efficiency of explorations, we develop a sampling distribution that can make the collected data give more feedbacks to the auto-bidding policy being trained. The second main algorithm is an *effective and stable* offline RL method to train the auto-bidding policy based on the collected data, named as the variance-suppressed conservative Q-learning (V-CQL). Specifically, motivated by the observation that the optimal[4] Q function is in the quadratic form, we design a regularization term in the V-CQL to optimize the shape of the Q function. The V-CQL can train the auto-bidding policy with high average — hence effective, and low variance — hence stable, in performance under different random seeds. The whole SORL works in an iterative manner, alternating between collecting data and training the auto-bidding policy. Extensive simulated and real-world experiments validate the effectiveness of our approach.

## 2   Related Work

**VAS-based RL Auto-bidding Methods.** Impressed by powerful contextual learning and sequential decision-making capabilities of RL, modern auto-bidding policies, such as DRLB [1], RLB [4], MSBCB [2] and USCB (state-of-the-art) [5], are usually learned by RL algorithms in a manually built VAS. However, as stated before, they all suffer from the IBOO problem. The SORL avoids using the VAS and thereby completely address the IBOO challenge.

**Offline RL.** Offline RL (also known as batch RL) [11, 13, 26, 27, 30, 31, 38, 39, 40] aims to learn better policies based on a fixed offline dataset collected by some behavior policies. The main challenge offline RL addressed is the extrapolation error caused by missing data [27]. Specifically, offline RL algorithms usually address this challenge in three ways, including policy constraint methods such as BCQ [11], BEAR [30], and conservative regularization methods such as CQL [13], BRAC [40], as well as modifications of imitation learning [37] such as ABM [31], CRR [38], BAIL[39]. However, there is another important challenge that offline RL cannot solve: the fixed offline dataset cannot be guaranteed to contain sufficient state-action pairs from high-reward regions [27]. This challenge exists in many practical applications, including the auto-bidding problem, and can inherently prevent offline RL algorithms from learning excellent policies. Hence, a great advantage of the SORL is its ability to continuously collect data from OOD high-reward regions that can give new feedbacks to the auto-bidding policy being trained.

**Off-Policy Evaluation (OPE).** OPE is an algorithm to evaluate the performance of the policy with offline data collected by other policies [12]. The state-of-the-art OPE algorithm in auto-bidding is to calculate the $R/R*$ of the evaluated policy in the VAS (see Appendix B for details). However, as shown in Table. 1, the OPE in auto-bidding is not accurate. This indicates that we cannot rely on the OPE to select auto-bidding policies with good performance for directing further online explorations. Notably, the proposed offline training algorithm in the SORL, V-CQL, outperforms existing offline RL algorithms in training stabilities and helps to reduce the OPE process during iterations in SORL.

**Online RL.** Safety is of vital importance in the online exploration of many real-world industrial systems, including the RAS. Many safe online RL algorithms have been proposed to guarantee the safety of explorations [17, 18, 19, 20, 21, 22, 33, 34, 35]. However, many of them are either developed based on the constraints that are not suitable for the auto-bidding problem or designed for systems with specific assumptions (see Appendix B for details). Besides, many existing works [33, 34, 35, 21] assume that there exists a specific set of safe state-action pairs that can be explored. Some work [33] even requires to know the safe states in advance. However, in the auto-bidding problem, no specific actions at any state cannot be explored as long as the expected accumulative reward maintains at a high level. This requires the exploration policy to maintain high performance throughout the iterations, which is more challenging. Notably, the SER policy can meet this requirement with theoretical guarantees. Recently, with the development of the offline RL field, many algorithms for efficient online explorations on the premise of having an offline dataset [14, 23] have emerged. However, they often only focus on the efficiency of data collections but ignore the safety of it. Notably, the SER policy in the SORL strikes a good balance between the efficiency and safety.

---

[4]Here "optimal" refers to the optimal Q function in the simulated experiments. See Section 4.2 for details.

Supplementary related works are described in Appendix B.

# 3  Problem Settings

We consider the auto-bidding problem from the perspective of a single advertiser, which can be viewed as an episodic task with $T \in \mathbb{N}_+$ time steps. Specifically, between time step $t$ and $t + 1$, there are $N_t \in \mathbb{N}_+$ impression opportunities, each of which has a positive value $v_{j,t} \leq v_M$ and is sold to the winning advertiser at a market price $p_{j,t} \leq p_M$, where $v_M > 0$ and $p_M > 0$ are the upper bounds for values and market prices, respectively, $j \in \{1, 2, ..., N_t\}, t \in \{1, 2, ..., T\}$. Denote $p_{j,t}^1 \leq p_M$ and $v_{j,t}^1 \leq v_M$ as the market price and rough value of impression opportunity $j$ in stage 1, and $p_{j,t}^2 \leq p_M$ and $v_{j,t}^2 \leq v_M$ as the market price and accurate value of it in stage 2. Note that they are all positive values. Let the total budget of the advertiser be $B > 0$. The auto-bidding problem can be modeled as a Constraint Markov Decision Process (CMDP) [15] $< \mathcal{S}, \mathcal{A}, R, \mathbb{P}, \gamma, C >$. The state $s_t \triangleq [b_t, T - t, B - b_t] \in \mathcal{S}$ is composed of three elements[5], including the budget left $s_t(1) = b_t$, the time left $s_t(2) = T - t$ and the consumed budget $s_t(3) = B - b_t$. The action $a_t \in \mathcal{A} \triangleq [A_{\min}, A_{\max}]$ is the "a-bid" at time step $t$, where $A_{\max} > A_{\min} > 0$ are the upper and lower bounds, respectively. The real bidding price for impression opportunity $j$ under action $a_t$ is $a_t v_{j,t}$. See [5] for detailed explanations. In addition, the reward function $r_t(s_t, a_t)$ and the constraint function $c_t(s_t, a_t)$ are the total value of impression opportunities won between time step $t$ and $t+1$ and the corresponding costs, respectively, and $\mathbb{P}$ denotes the state transition rule. Note that $R$, $C$ and $\mathbb{P}$ are all directly determined by the RAS. Moreover, $\gamma \in [0, 1]$ is the discounted factor. We denote the auto-bidding policy as $\mu : \mathcal{S} \to \mathcal{A}$, and let $\Pi$ be the policy space. The goal of the auto-bidding problem is to maximize the total values of the impression opportunities earned by the advertiser under the budget constraint, which can be expressed as

$$\max_{\mu \in \Pi} V(\mu) \triangleq \mathbb{E}_{s_{t+1} \sim \mathbb{P}, a_t \sim \mu} \left[ \sum_{t=0}^{T-1} \gamma^t r_t(s_t, a_t) \right], \quad \text{subject to} \quad \sum_{t=0}^{T-1} c_t(s_t, a_t) \leq B. \tag{1}$$

One can leverage standard RL algorithms to train a state-of-the-art auto-bidding policy [5] with the VAS, where the constraint in (1) is met by terminating the training episode early once the budget runs out. However, as stated before, this way of learning suffers from the challenge of IBOO. Hence, we propose the SORL framework to avoid the IBOO in the following section.

# 4  Our Approach

The SORL framework consists of two algorithms, including the SER policy for online explorations and data collections, as well as the V-CQL method for offline training of the auto-bidding policy based on the collected data. The SORL works in an iterative manner, alternating between online data collections and offline training.

## 4.1  Online Exploration: SER Policy

There are two requirements on the online exploration policy. As the exploration policy is directly deployed in the RAS, the primary requirement is *safety*. Unlike the safety in other realms such as robotics, it is not appropriate to construct an *immediate-evaluated* safety function to assess the safety degree of each state-action pair merely based on their values in auto-bidding. Actually, any action in any state would be safe as long as the summation of all rewards of the whole episode maintains at a high level. See Appendix F.1.3 for detailed explanations. Let $\mu_s$ be the auto-bidding policy originally[6] deployed in the RAS. Note that $\mu_s$ is safe. Denote the online exploration policy as $\pi_e$. We can formally define the safety requirement as: *the performance of $\pi_e$ cannot be lower than that of*

---

[5]In fact, more elements can be added to the state, and we claim that the under mild additional assumptions, the propositions and theorems in this paper still hold.

[6]There is always an auto-bidding policy $\mu_s$ deployed in the RAS. For example, the state-of-the-art method USCB [5] will do.

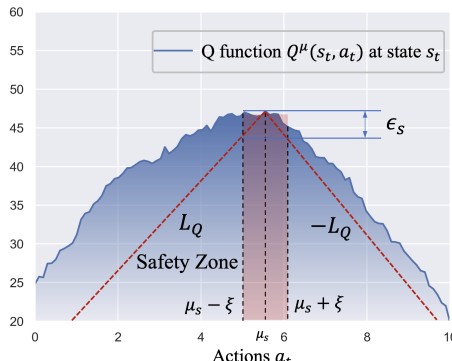
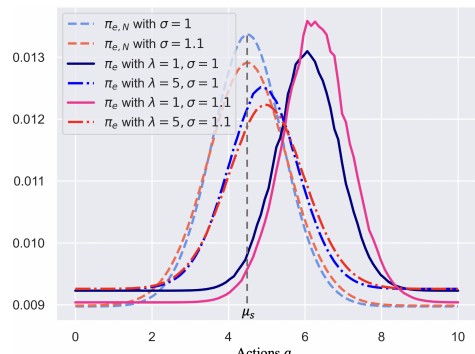

(a) The Q function of any auto-bidding policy is $L_Q$-Lipschitz smooth. The safety zone is designed as the range between $\mu_s(s_t) - \xi$ and $\mu_s(s_t) + \xi$, where $\xi \leq \frac{\epsilon_s}{L_Q \gamma^{t_1} \Delta T}$.

(b) Sampling distribution of the SER policy, $\tilde{\pi}_e$, under different hyper-parameters $\sigma$ and $\lambda$, as well as $\tilde{\pi}_{e,\mathcal{N}}$ with different hyper-parameter $\sigma$.

Figure 1: Design of the safety zone and sampling distributions in the SER policy.

$\mu_s$ by a threshold $\epsilon_s > 0$, i.e., $V(\mu_s) - V(\pi_e) \leq \epsilon_s$. Besides, $\pi_e$ needs to be *efficient*, i.e., *it should collect data that can give new feedbacks to the auto-bidding policy being trained*, which constitutes the second requirement. We next propose the SER policy for online explorations that can satisfy both requirements.

### 4.1.1 Theory: Lipschitz Q Function

Our basic idea of ensuring the safety of exploration is to design a safety zone for taking actions around the outputs of $\mu_s$. The theoretical foundation of this idea is our proof of the Lipschitz smooth property of the Q function in the auto-bidding problem. We next report the theorem of this Lipschitz smooth property, as well as corresponding propositions and assumptions. Let $\mu$ be an arbitrary auto-bidding policy. According to the Bellman equation, we have:

$$Q^\mu(s_t, a_t) = r_t(s_t, a_t) + \gamma \mathbb{E}_{s_{t+1} \sim \mathbb{P}(\cdot|s_t, a_t)}[Q^\mu(s_{t+1}, \mu(s_{t+1}))]. \tag{2}$$

Based on the mechanism of RAS, we formulate the reward function $R$, the constraint function $C$ and the transition rule $\mathbb{P}$ as follows, and the proof can be found in Appendix D.1.

**Proposition 1** (Analytical expressions of $R$, $C$ and $\mathbb{P}$). *Based on the characteristic of the two-stage cascaded auction in the RAS, we can formulate the reward function $R$ as $r_t(s_t, a_t) = \sum_j \mathbb{1}\{a_t v_{j,t}^1 \geq p_{j,t}^1, a_t v_{j,t}^2 \geq p_{j,t}^2\} v_{j,t}$, the constraint function $C$ as $c_t(s_t, a_t) = \sum_j \mathbb{1}\{a_t v_{j,t}^1 \geq p_{j,t}^1, a_t v_{j,t}^2 \geq p_{j,t}^2\} p_{j,t}$, and the state transition rule $\mathbb{P}$ as $s_{t+1} = s_t + [\triangle s_t(1), \triangle s_t(2), \triangle s_t(3)]$, where $\triangle s_t(1) = -\triangle s_t(3) = -\sum_j \mathbb{1}\{a_t v_{j,t}^1 \geq p_{j,t}^1, a_t v_{j,t}^2 \geq p_{j,t}^2\} p_{j,t}$ and $\triangle s_t(2) = -1$.*

We make the following assumption on the impression opportunities, whose rationalities can be found in Appendix C.1.

**Assumption 1** (Bounded Impression Distributions). *Between time step $t$ and $t + 1$, we assume the numbers of winning impressions with action $a_t$ in the first stage $n_{t,1}$ and the second stage $n_{t,2}$ can both be bounded by linear functions, i.e., $n_{t,1} \leq k_1 a_t, n_{t,2} \leq k_2 a_t$, where $k_1, k_2 > 0$ are constants.*

Based on this assumption, we claim that the reward function $r_t(s_t, a_t)$ is Lipschitz smooth with respect to actions $a_t$ at any given state $s_t$. See Appendix E.1 for the proof.

**Theorem 1** (Lipschitz Smooth of $r_t(s_t, a_t)$). *Under Assumption 1, the reward function $r_t(s_t, a_t)$ is $L_r$-Lipschitz smooth with respect to actions $a_t$ at any given state $s_t$, where $L_r = (k_1 + k_2) v_M$.*

We make the following mild assumptions, whose rationalities can be found in Appendix C.2.

**Assumption 2** (Bounded Partial Derivations of $Q^\mu(s_t, a_t)$). *We assume that the partial derivation of $Q^\mu(s_t, a_t)$ with respect to $s_t(1)$ and $s_t(3)$ is bounded, i.e., $\left|\frac{\partial Q^\mu(s_t, a_t)}{\partial s_t(1)}\right| \leq k_3$ and $\left|\frac{\partial Q^\mu(s_t, a_t)}{\partial s_t(3)}\right| \leq k_4$, where $k_3, k_4 > 0$ are constants.*

Equipped with Theorem 1, we can prove that the Q function $Q^\mu(s_t, a_t)$ is also Lipschitz smooth. See Appendix E.2 for proof.

**Theorem 2** (Lipschitz Smooth of $Q^\mu(s_t, a_t)$). *Under Assumption 1 and 2, the Q function $Q^\mu(s_t, a_t)$ is an $L_Q$-Lipschitz smooth function with respect to the actions $a_t$ at any given state $s_t$, where $L_Q = [v_M + \gamma(k_3 + k_4)p_M](k_1 + k_2)$.*

This means that the decrease rate of $Q^\mu$, the subsequent accumulated rewards, due to action offset at any time step $t$ is bounded by $L_Q$, which gives us a way to design the safety zone. Specifically, as shown in Fig. 1(a), the safety zone at state $s_t$ can be design as the neighborhood of action $\mu_s(s_t)$, i.e., $[\mu_s(s_t) - \xi, \mu_s(s_t) + \xi]$, where $\xi > 0$ is the range. In this way, the online exploration policy $\pi_e$ can be designed as: sampling within the safety zone $[\mu_s(s_t) - \xi, \mu_s(s_t) + \xi]$ in certain $\Delta T \geq 1$ consecutive time steps, and sticking to $\mu_s$ in the rest of $T - \Delta T$ time steps, i.e.,

$$\pi_e(s_t) = \begin{cases} \text{sampling from } [\mu_s(s_t) - \xi, \mu_s(s_t) + \xi], & t_1 \leq t \leq t_2; \\ \mu_s(s_t), & \text{otherwise.} \end{cases} \quad \forall t, \tag{3}$$

where $0 \leq t_1 < t_2 \leq T - 1$, $\Delta T = t_2 - t_1 + 1$, and $t_1, t_2 \in \mathbb{N}^+$. In the following theorem, we give the upper bound of $|V(\pi_e) - V(\mu_s)|$.

**Theorem 3** (Upper Bound of $|V(\pi_e) - V(\mu_s)|$). *The expected accumulated reward $V(\pi_e)$ satisfies*

$$\left| V(\pi_e) - V(\mu_s) \right| \leq \xi \gamma^{t_1} \left[ v_M + \gamma(k_3 + k_4)p_M \right] (k_1 + k_2) \Delta T. \tag{4}$$

See Appendix E.3 for the detailed proofs. Hence, to meet the safety requirement, we design the range of the safety zone as $\xi \leq \frac{\epsilon_s}{L_Q \gamma^{t_1} \Delta T}$.

### 4.1.2 Sampling Method

With the safety guarantee, we can further design the sampling method in $\pi_e$ to increase the efficiency of exploration.

**Vanilla Sampling Method.** A vanilla sampling method in the safety zone can directly be sampling from a (truncated) Gaussian distribution $\tilde{\pi}_{e,\mathcal{N}} = \mathcal{N}(\mu_s(s_t), \sigma^2)$ with mean $\mu_s(s_t)$ and variance $\sigma^2$. However, $\tilde{\pi}_{e,\mathcal{N}}$ resembles the policy $\mu_s$ to a large extent and cannot change with the auto-bidding policy being trained, which makes the explorations conservative and lack of new feedbacks.

**SER Sampling Method.** To lift up the efficiency of explorations, we shift the distribution $\tilde{\pi}_{e,\mathcal{N}}$ towards the actions of the auto-bidding policy being trained, while constraining the deviations within a threshold $\epsilon_e > 0$. This can be formulated as a functional optimization problem:

$$\max_{\tilde{\pi}_e, \forall s_t} \quad \mathbb{E}_{a_t \sim \tilde{\pi}_e(\cdot|s_t)} \widehat{Q}(s_t, a_t) \qquad \text{s.t.} \quad D_{\text{KL}}(\tilde{\pi}_e, \tilde{\pi}_{e,\mathcal{N}}) \leq \epsilon_e, \tag{5}$$

where the optimization variable $\tilde{\pi}_e$ denotes the shifted distribution, $\widehat{Q}$ is the Q function of the auto-bidding policy being trained. Using Euler equations, we can derive the form of $\tilde{\pi}_e$ from (5) as

$$\tilde{\pi}_e = \frac{\tilde{\pi}_{e,\mathcal{N}}}{C(s_t)} \exp\left\{ \frac{1}{\lambda} \widehat{Q}(s_t, a_t) \right\} = \frac{1}{C(s_t)} \exp\left\{ \underbrace{-\frac{(a_t - \mu_s(s_t))^2}{2\sigma^2}}_{\text{safety}} + \underbrace{\frac{1}{\lambda} \widehat{Q}(s_t, a_t)}_{\text{efficiency}} \right\}, \tag{6}$$

where $C(s_t) = \int_{a_t} \exp\{-\frac{(a_t - \mu_s(s_t))^2}{2\sigma^2} + \frac{1}{\lambda}\widehat{Q}(s_t, a_t)\} da_t$ acts as the normalization factor. The complete deductions of $\tilde{\pi}_e$ are given in Appendix F.1.1. We can interpret $\tilde{\pi}_e$ as a deviation from $\tilde{\pi}_{e,\mathcal{N}}$ towards the Q function $\widehat{Q}(s_t, a_t)$, where $\tilde{\pi}_{e,\mathcal{N}}$ further guarantees the safety and the Q function ensures the efficiency. Note that $\sigma$ and $\lambda$ in (6) are both hyper-parameters that can control the randomness and deviation degrees from $\mu_s$ of the SER policy, respectively. As shown in Fig. 1(b), the smaller the value of $\lambda$, the larger the deviation degree of $\tilde{\pi}_e$ from $\mu_s$; besides, the bigger the value of $\sigma$, the greater the randomness degree of $\tilde{\pi}_e$. Hence, we can control the randomness and deviation degrees from the safe policy $\mu_s$ of exploration policy easily by adjusting the value of $\sigma$ and $\lambda$. In addition, we describe the way of practically implementing the sampling of distribution $\tilde{\pi}_e$ in Appendix F.1.2. Then the complete SER policy is

$$\pi_e(s_t) = \begin{cases} \text{sampling from } [\mu_s(s_t) - \xi, \mu_s(s_t) + \xi] \text{ with } \tilde{\pi}_e, & t_1 \leq t \leq t_2; \\ \mu_s(s_t), & \text{otherwise.} \end{cases} \quad \forall t. \tag{7}$$

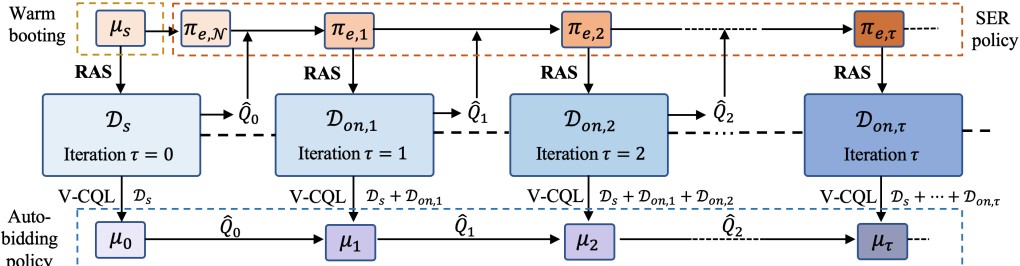

Figure 2: Sustainable online reinforcement learning (SORL) framework which learns the auto-bidding policy directly with the RAS in a safe and efficient way. The SORL works in an iterative manner.

## 4.2 Offline Training: V-CQL

Due to the safety constraint on the explorations, the SER policy can only collect data within the safety zone and the data outside the safety zone will be missed. Hence, we leverage a strong baseline offline RL algorithm, CQL, to address the extrapolation error and make the offline training *effective*. Moreover, as the sampling distribution $\tilde{\pi}_e$ in the SER policy involves the Q function $\widehat{Q}$, the training result will affect the directions of further explorations. However, on the one hand, as stated in Section 2, the OPE in the auto-bidding is not reliable; and on the other hand, the performance variance under different random seeds of existing offline RL algorithms can be large, as shown in Fig. 5.2 in the experiment. Hence, we further design a novel regularization term in the CQL loss to increase the *stability* of offline training and thereby reduce the OPE process. This forms the V-CQL method. Specifically, we observe that the Q functions of the state-of-the-art auto-bidding policy are always in nearly quadratic forms. In addition, we conduct simulated experiments where the auto-bidding policy is directly trained in a simulated RAS[7] with traditional RL algorithms. We find out the optimal Q functions in the simulated experiments are also in nearly quadratic forms. See Appendix F.2.1 for illustrations. Based on these observations, we assume that the optimal Q functions in the RAS are also in nearly quadratic forms. Hence, the key idea of V-CQL is to restrict the Q function to remain nearly quadratic, and the regularization term in the V-CQL is designed as:

$$\mathcal{R}(\mu) = \beta \mathbb{E}_{s_k \sim \mathcal{D}_s} \left[ D_{\mathrm{KL}} \left( \underbrace{\frac{\exp(\widehat{Q}(s_k, \cdot))}{\sum_a \exp(\widehat{Q}(s_k, a))}}_{\text{distribution(form) of } \widehat{Q}}, \underbrace{\frac{\exp(\widehat{Q}_{\mathrm{qua}}(s_k, \cdot))}{\sum_a \exp(\widehat{Q}_{\mathrm{qua}}(s_k, a))}}_{\text{distribution(form) of } \widehat{Q}_{\mathrm{qua}}} \right) \right], \tag{8}$$

where $Q_{\mathrm{qua}}$ is selected as the Q function of the state-of-the-art auto-bidding policy, $D_{\mathrm{KL}}(\cdot, \cdot)$ represents the KL-divergence, and $\beta > 0$ is a constant controlling the weight of $\mathcal{R}(\mu)$. Moreover, we can interpret (8) from another aspect, i.e., $\mathcal{R}(\mu)$ tries to limit the distance between the distribution of the Q function and the distribution of $\widehat{Q}_{\mathrm{qua}}$. This can limit the derivation of the auto-bidding policy from the corresponding state-of-the-art policy, which can increase the training stability. Complete implementations of the V-CQL are shown in Appendix F.2.2.

## 4.3 Iterative Updated Structure

Fig. 2 shows the whole structure of the SORL framework. Specifically, the SORL works in an iterative manner, alternating between collecting data with the SER policy from the RAS and training the auto-bidding policy with the V-CQL method.

**Warm Booting.** To start with, we use the safe policy $\mu_s$ to boot the explorations in the RAS. Denote the data collected by $\mu_s$ as $\mathcal{D}_s = \{(s_k, a_k, r_k, s'_k)\}_k$, and the auto-bidding policy trained by $\mathcal{D}_s$ with the V-CQL as $\mu_0$.

**Iteration Process.** Denote the SER policy and the auto-bidding policy in the $\tau$-th iteration as $\pi_{e,\tau}$ and $\mu_\tau$, and the data collected in the $\tau$-th iteration as $\mathcal{D}_{on,\tau}$, where $\tau \in \{1, 2, ...\}$ The design

---

[7]The details of this simulated RAS are described in Appendix A.2

for the sampling distribution $\tilde{\pi}_{e,\tau}$ in iteration $\tau$ is $\tilde{\pi}_{e,\tau} = \frac{1}{C_\tau(s_t)}\tilde{\pi}_{e,\mathcal{N}}\exp\{\widehat{Q}_\tau(s_t, a_t)/\lambda_\tau\}$, where $C_\tau(s_t)$ acts as the normalization factor and $\lambda_\tau$ is the hyper-parameter used in iteration $\tau$, and $\widehat{Q}_\tau$ is the Q function of $\mu_\tau$. We note that the exploration policies in each iteration are safe, i.e., $|V(\pi_{e,\tau}) - V(\mu_s)| \leq \epsilon_s, \forall \tau$, since they all taking actions within the safety zone only with different sampling distributions. We leverage the V-CQL to in each iteration improve the auto-bidding policy. Specifically, at iteration $\tau$, we substitute $\widehat{Q}_{\text{qua}}$ in (8) with $\widehat{Q}_{\tau-1}$, and train a new Q function $\widehat{Q} \leftarrow \widehat{Q}_\tau$ for the next iteration. The auto-bidding policy are expected to be continuously improved. A summary of the SORL framework is present in Appendix F.3.

# 5   Experiments

We conduct both simulated and real-world experiments to validate the effectiveness of our approach[8]. The following three questions are mainly studied in the experiments: (1) What is the performance of the whole SORL framework? Is the SER policy safe during iterations? Can the V-CQL method continuously improve the auto-bidding policy and outperform existing offline RL algorithms and the state-of-the-art auto-bidding policy? (2) Is the safety zone reliable? Is the SER policy still safe when using Q functions of auto-bidding policies with bad performance? (3) Does the V-CQL really help to reduce the performance variance compared to existing offline RL algorithms?

**Experiment Setup.** We conduct the real-world experiments on one of the world's largest E-commerce platforms, TaoBao. See Appendix G.1 for details. The simulated experiments are conducted in a manually built offline RAS and the corresponding VAS. See Appendix A.2 for details. The safe auto-bidding policy $\mu_s$ used for warm booting and constructing the safety zone is trained by the state-of-the-art auto-bidding policy, USCB[5]. The safety threshold is set as $\epsilon_s = 5\%V(\mu_s)$.

**Performance Index.** The objective function $V(\mu)$ in (1), i.e., the total value of impression opportunities won by the advertiser in the episode, acts as the main performance index in our experiments and is referred as *BuyCnt* in the following. In addition, we utilize three other metrics that are commonly used in the auto-bidding problem to evaluate the performance of our approach. The first metric the total *consumed budget (ConBdg)* of the advertiser. The second metric is the *return on investment (ROI)* which is defined as the ratio between the total revenue and the ConBdg of the advertiser. The third metric is the *cost per acquisition (CPA)* which is defined as the average cost for each successfully converted impression. Note that larger values of BuyCnt, ROI, and ConBdg with a smaller value of CPA indicate better performance of the auto-bidding policy.

**Baselines.** We compare our approach with the state-of-the-art auto-bidding policy, USCB [5], that is trained by RL in the VAS. We also compare the V-CQL method with modern offline RL algorithms, including BCQ [11] and CQL [13]. Recall that many safe online RL algorithms are not suitable for explorations in the auto-bidding problem as stated in Section 2, we compare the SER policy with $\pi_{e,\mathcal{N}}$ in our experiments.

## 5.1   Main Results

**To Answer Question (1):** We first conduct simulated experiments with the SORL, and the results during iterations are shown in Fig. 5.1. Specifically, from Fig. 3(a), we can see that the decline rate in BuyCnt of both the SER policy and the vanilla exploration policy are smaller than $5\%$, which verifies the safety of the SER policy. Moreover, the BuyCnt of the SER policy $\pi_{e,\tau}$ rises with iterations and is alway higher than that of $\pi_{e,\mathcal{N}}$, which indicates that the SER policy is more efficient. From Fig. 3(b), we can see that the BuyCnt of the auto-bidding policy $\mu_\tau$ trained with V-CQL is higher than that of BCQ, CQL and USCB, which indicates the effectiveness of the V-CQL. Besides, the BuyCnt rises with the number of iterations and converges to the optimal BuyCnt (i.e., the BuyCnt of the optimal policy) at the 5-th iteration. This validates the superiority of the whole SORL framework. This indicates the effectiveness of the V-CQL method. For real-world experiments, we utilize $10,000$ advertisers to collect data from the RAS with $\pi_e$ and compare the auto-bidding policies in $4$ iterations on $1,500$ advertisers using A/B tests, and the results are shown in Table. 2. We can see that the performances of auto-bidding policies are getting better with iterations and exceeds the state-of-the-art algorithm, USCB, which validates the superiority of the whole SORL framework.

---

[8]The codes of simulated experiments are available at https://github.com/nobodymx/SORL-for-Auto-bidding.

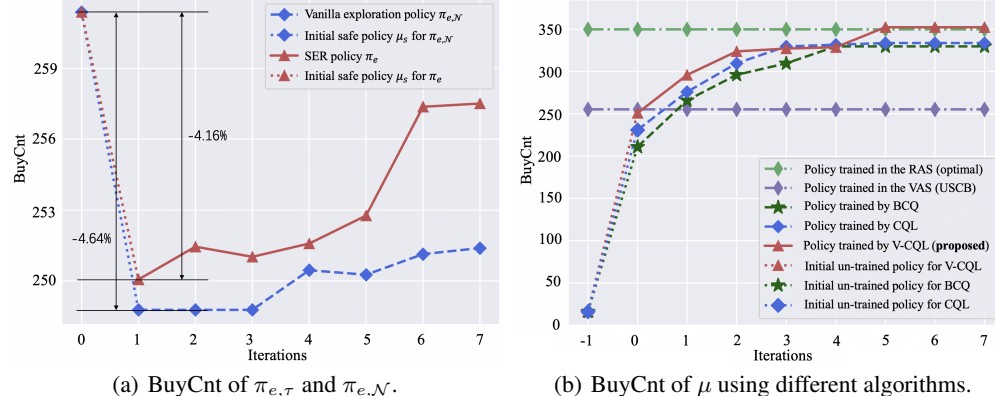

(a) BuyCnt of $\pi_{e,\tau}$ and $\pi_{e,\mathcal{N}}$.  (b) BuyCnt of $\mu$ using different algorithms.

Figure 3: The change of BuyCnt of both the SER policy $\pi_{e,\tau}$ and auto-bidding policy $\mu_\tau$ with iterations $\tau$ when applying the SORL framework.

Table 2: The results of SORL framework in the real-world experiments.

| Iterations | A/B Tests with $\mu_{\tau-1}$ | | | | A/B Tests with the safe policy $\mu_s$ | | | |
|---|---|---|---|---|---|---|---|---|
| | BuyCnt | ROI | CPA | ConBdg | BuyCnt | ROI | CPA | ConBdg |
| 0-th: $\mu_0$ | **+3.21%** | **+1.28%** | **-2.01%** | **+1.12%** | **+3.21%** | **+1.28%** | **-2.01%** | **+1.12%** |
| 1-th: $\mu_1$ | **+0.65%** | **+1.96%** | **-1.27%** | -0.62% | **+3.41%** | **+2.88%** | **-0.93%** | **+2.45%** |
| 2-th: $\mu_2$ | **+0.47%** | **+0.26%** | **-0.13%** | **+0.33%** | **+3.57%** | **+1.60%** | **-0.98%** | **+2.55%** |
| 3-th: $\mu_3$ | **+0.95%** | **+3.20%** | **-1.01%** | **+0.06%** | **+3.75%** | **+2.48%** | **-3.91%** | -0.15% |

## 5.2 Ablation Study

**To Answer Question (2):** We fully examine the safety of the SER policy using the Q function $\widehat{Q}$ of auto-bidding policies with different performance levels. Specifically, in the simulated experiment, we utilize seven different versions of auto-bidding policies $\{\mu_{(k)}\}_{k=1}^{7}$ with Q functions $\{\widehat{Q}_{(k)}\}_{k=1}^{7}$ that have various performance levels to construct seven corresponding SER policies, where $V(\mu_{(1)}) < V(\mu_{(2)}) < ..., V(\mu_{(7)})$. We also construct a vanilla exploration policy $\pi_{e,\mathcal{N}}$ for comparison. The hyper-parameters for all exploration policies are $\sigma = 1, \lambda = 0.1$. We apply the $\pi_{e,\mathcal{N}}$ and seven SER policies to the simulated RAS, and the BuyCnt of explorations policies are shown in Fig. 4. We can

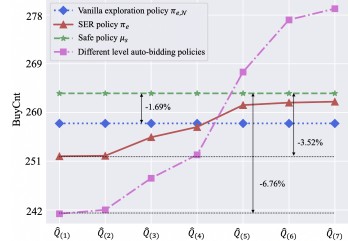

Figure 4: BuyCnt of $\pi_{e,\mathcal{N}}, \pi_e$.

see that the BuyCnt of the SER policy rises with the performance level of the auto-bidding policy. The worst BuyCnt of the SER policy drops about $3.52\%$ compared with $\mu_s$, which meets the safety requirement. Moreover, we can balance between the safety and efficiency of the SER policy by regulating the hyper-parameters $\sigma$ and $\lambda$. The corresponding results are shown in Appendix G.2.2.

For real-world experiments, we apply $\pi_e$ and $\pi_{e,\mathcal{N}}$ to 1, 300 advertisers in the RAS and compare their performance to $\mu_s$ in A/B tests, as shown in Table. 3. We can see that the variations of the ConBdg and BuyCnt of $\pi_e$ are all with in $5\%$, and are better than those of $\pi_{e,\mathcal{N}}$, which indicates the safety of the SER policy.

Table 3: Real-world A/B tests between $\pi_e$ and $\mu_s$, as well as between $\pi_{e,\mathcal{N}}$ and $\mu_s$.

| Methods | BuyCnt | ConBdg |
|---|---|---|
| vanilla $\pi_{e,\mathcal{N}}$ | -3.32% | +1.11% |
| SER policy $\pi_e$ | **-1.83%** | **+0.67%** |

**To Answer Question (3):** We leverage the V-CQL, CQL, BCQ and USCB to train auto-bidding policies under 100 different random seeds in the simulated experiment, and the results are shown in Fig. 5(a). We can see that the performance variance of the V-CQL is much smaller than those of other algorithms. At the same time, the average performance of the V-CQL can maintains at a high level. In addition, we leverage the V-CQL and CQL algorithm to train the auto-bidding policies based on the real-world data under 100 different random seeds. To ensure fairness, we also utilize the USCB to train the auto-bidding policies in the VAS under exactly the same random seeds. We carry out the OPE for these three sets of auto-bidding policies, as shown in Fig. 5(b). Recall that $R/R*$ is the main

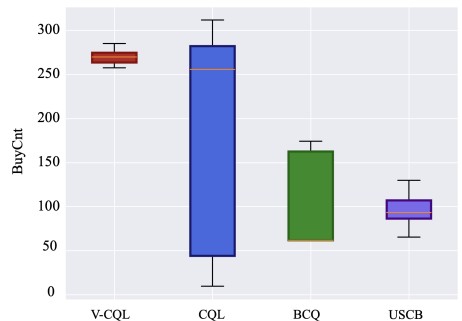
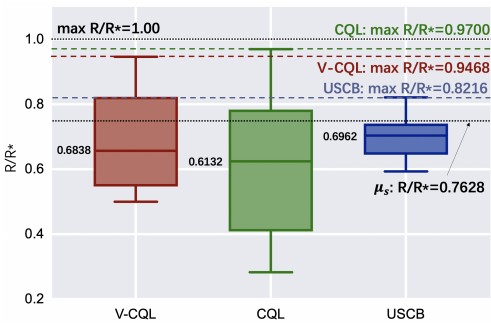

(a) BuyCnt of the auto-bidding policy trained by different methods in the simulated RAS.

(b) OPE of the auto-bidding policy trained by different methods with real-world data.

Figure 5: The performance of different methods under 100 random seeds.

main metric used in the OPE for auto-bidding (see Appendix B). We can see that the maximum $R/R^*$ of the V-CQL is much larger than that of the USCB, which indicates that the V-CQL is capable of training a better policy. Although the maximum $R/R*$ of the V-CQL and the CQL are about in the same level, the variance of $R/R*$ of the V-CQL is smaller than that of the CQL. For real-world experiments, we apply the auto-bidding policies trained by the V-CQL and USCB to $1,600$ advertisers in the A/B test for seven days in the RAS, and the average values of the metrics are

Table 4: Real-world A/B tests between the V-CQL and USCB under different random seeds.

| Seeds | BuyCnt | Seeds | BuyCnt |
|---|---|---|---|
| 1 | **+1.70%** | 6 | **+1.94%** |
| 2 | **+1.06%** | 7 | **+1.38%** |
| 3 | **+3.01%** | 8 | **+1.19%** |
| 4 | **+1.89%** | 9 | **+0.31%** |
| 5 | -0.53% | 10 | **+0.71%** |

shown in Table 5. We can see that the V-CQL outperforms USCB method in almost all metrics. Moreover, we present the A/B test results under 10 random seeds in Table. 4. We can see that the V-CQL outperforms the USCB under 9 seeds, and only slightly worse under the other one seed. All these results indicate that the V-CQL can really help to reduce the performance variance and increase the training stability, while keeping the average performance at a high level.

Table 5: Real-world A/B Tests between the V-CQL and USCB.

| Methods | USCB: policy trained with new data | | | | USCB : the safe policy $\mu_s$ | | | |
|---|---|---|---|---|---|---|---|---|
| | BuyCnt | ROI | CPA | ConBdg | BuyCnt | ROI | CPA | ConBdg |
| USCB | 40,926 | 3.90 | 20.71 | 847,403.12 | 35,627 | 3.82 | 21.58 | **768,832.64** |
| V-CQL | **42,236** | **3.95** | **20.29** | **856,913.14** | **37,090** | **3.97** | **20.61** | 764,467.39 |
| variations | **+3.20%** | **+1.28%** | **-2.03%** | **+1.12%** | **+4.11%** | **+3.93%** | **-4.49%** | -0.57% |

# 6 Conclusions

In this paper, we study the auto-bidding problem in online advertisings. Firstly, we define the IBOO challenge in the auto-bidding problem and systematically analyze its causes and influences. Then, to avoid the IBOO, we propose the SORL framework that can directly learn the auto-bidding policy with the RAS. Specifically, the SORL framework contains two main algorithms, including a safe and efficient online exploration policy, the SER policy, and an effective and stable offline training method, the V-CQL method. The whole SORL framework works in an iterative manner, alternating between online explorations and offline training. Both simulated and real-world experiments validate the superiority of the whole SORL framework over the state-of-the-art auto-bidding algorithm. Moreover, the ablation study shows that the SER policy can guarantee the safety of explorations even under the guide of the auto-bidding policy with bad performance. The ablation study also verifies the stability of the V-CQL method under different random seeds.

## Acknowledgments and Disclosure of Funding

This work is supported by Alibaba Research Intern Program. The authors would like to thank Mingyuan Cheng, Guan Wang, Zongtao Liu, Zhaoqing Peng, Lvyin Niu, Miao Xu, and Tianyu Wang for their valuable feedbacks and insightful discussions.

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
