# Outline of the Supplementary Material

The Appendix is mainly organized as follows[1].

- Appendix A presents more information on the backgrounds, as well as a further discussion on the motivation of this paper, i.e., IBOO problem in the auto-bidding.
    - Appendix A.1 presents the detailed structures of the RAS and VAS.
    - Appendix A.2 presents the details of the simulated advertising system experiments shown in Fig. 2 in the manuscript.
    - Appendix A.3 presents the figures illustrating the causes and influence and IBOO.
    - Appendix A.4 discusses the importance and universality of the IBOO problem.
- Appendix B represents the additional related works.
- Appendix C, D and E provides the theoretical supports for the Lipschitz smooth property of the safety function.
    - Appendix C.1 and C.2 provide rationality explanations on Assumption 1 and 2, respectively.
    - Appendix D.1 provide proofs of Proposition 1, respectively.
    - Appendix E.1, E.2 and E.3 provide proofs of Theorem 1, 2 and 3, respectively.
- Appendix F presents the detailed interpretations, implementations and derivations of the SORL framework.
    - Appendix F.1 presents the derivations and practical implementations on the SER policy.
        - - Appendix F.1.1 provides the derivations of the SER policy.
        - - Appendix F.1.2 provides an implementation of the SER policy in practice.
        - - Appendix F.1.3 describes more on the safety requirement in auto-bidding.
    - Appendix F.2 presents the motivations of the design on the V-CQL, and the complete implementation on the V-CQL, as well as its interpretation and relations to previous works.
        - - Appendix F.2.1 presents the motivations of the design on the V-CQL.
        - - Appendix F.2.2 presents the complete implementation on the V-CQL, as well as its interpretation and relations to previous works.
    - Appendix F.3 shows the pseudocode of the SORL framework.
- Appendix G presents the additional settings and results of the experiments.
    - Appendix G.1 shows the experiment setup.
    - Appendix G.2 presents the additional experiment results to validate the effectiveness of the proposed SORL algorithm.
        - - Appendix G.2.1 compares our V-CQL method to more popular offline RL algorithms, including, BCQ, CQL($\rho$), etc., which can act as an ablation study.
        - - Appendix G.2.2 shows the effect of hyper-parameters $\sigma$ and $\lambda$ on the SER policy.
        - - Appendix G.2.3 shows the detailed experiment data on the A/B test of the SORL framework.
        - - Appendix G.2.4 shows the comparison between our approach and the multi-agent auto-bidding algorithm.
- Appendix H presents the broader impact of this paper.

---

[1]Figures, tables and formulas in the appendix have all been renumbered. Unless specifically stated "in the manuscript", Fig. x, Table. x and (x) refer to the figure, table and formula with number x in the appendix, respectively. Nonetheless, the numbers of assumptions, propositions and theorems in the appendix are the same as in the manuscript. The references in the appendix refers to the references presented in the manuscript.

# Appendix

## A    Backgrounds and Motivations

In this section, we provide more information on the application backgrounds, including the detailed structures of the RAS and VAS, the structures of the simulated advertising system. We also discuss the importance and universality of the IBOO problem in auto-bidding, which acts as the motivation of this work.

### A.1    Detailed Structures of the RAS and VAS

Fig. 1 shows the detailed structures of the RAS and VAS. Particularly, the VAS is built based on the historical data of advertisers during bidding in stage 2 of the RAS. The VAS can interact with any auto-bidding policies, while the RAS cannot due to safety concerns.

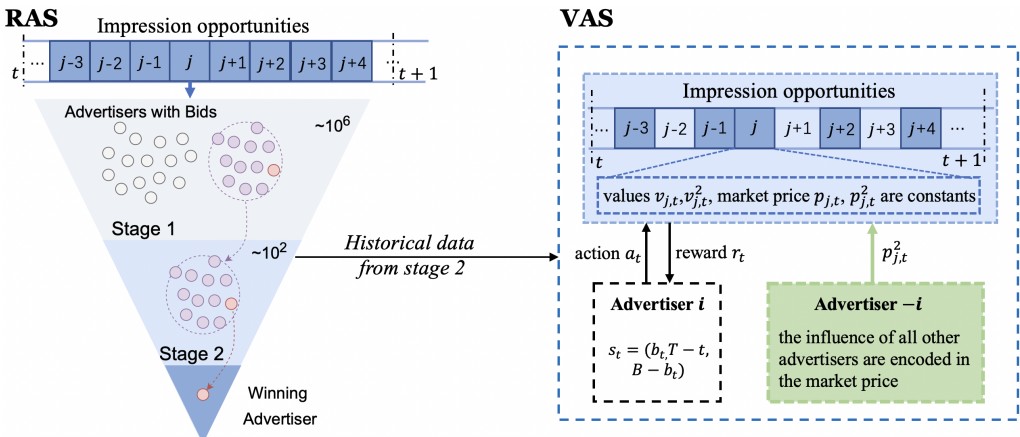

Figure 1: The RAS and VAS detailed structures, where the auction of an impression opportunity is completed in two stages in the RAS, and the advertiser in the VAS competes for each impression opportunity stored in the historical data from stage 2 with the stored market price.

#### A.1.1    Structures of RAS

In the RAS, the auction of each impression opportunity is completed through two stages. Specifically, consider an impression opportunity $j$ coming between time step $t$ and $t + 1$ with value $v_{j,t}$. All advertisers give the bids $a_{i,t}, i \in \mathbf{I}$ based on their current states, where $\mathbf{I}$ denotes the set of all advertisers. At stage 1, the RAS roughly calculates the value $v^1_{i,j,t}$ of impression opportunity $j$ with respect to each advertiser $i$, and compare the *effective cost per mile* (eCPM) value of all advertisers. The eCPM value of advertiser $i$ is defined as $a_{i,t}v^1_{i,j,t}$. A market price of impression opportunity $j$ is given by the RAS $p^1_{j,t}$ in stage 1, and advertisers with eCPM larger than $p^1_{j,t}$ can successfully enter the stage 2. Denote the set of advertisers entering stage 2 as $\mathbf{I}^2_j$. At stage 2, the RAS accurately evaluates the value $v^2_{i',j,t}$ of impression opportunity $j$ with respect to each advertiser $i' \in \mathbf{I}^2_j$. The advertiser with the largest eCPM $a_{i',t}v^2_{i',j,t}$ wins the impression $j$. Define the market price in stage 2 as the second highest eCPM among all advertisers, i.e.,

$$p^2_{j,t} \triangleq \max_{i' \in \mathbf{I}^2_j, i' \neq \arg\max_k a_{k,t}v^2_{k,j,t}} a_{i',t}v^2_{i',j,t}. \tag{1}$$

The winning advertiser earns the true value $v_{j,t}$ of impression $j$ and pays its market price given by the RAS $p_{j,t}$.

Note that we study the auto-bidding problem from the perspective of a single advertiser. Hence, in the manuscript (especially in Proposition 1), we omit the subscript $i$ and $i'$ in the values of impression opportunity $j$ in stage 1, $v_{j,t}^1$, and stage 2, $v_{j,t}^2$, respectively. We also omit the subscript $i$ in action $a_t$ in the manuscript.

### A.1.2   Structures of VAS

The VAS is built only based on the historical data of advertisers during bidding in stage 2 of the RAS. This is because the amount of data generated in stage 1 of the RAS is very large (about $10^2$ to $10^4$ impression opportunities coming to stage 1 at every moment, and about $10^6$ advertisers bidding for each impression), and it is computationally infeasible to build the VAS based on data in stage 1 of the RAS.

There exists some impression opportunities that are not in the VAS, such as impression opportunities $j-2$, $j+1$ and $j+3$ as shown in the VAS in Fig. 1. This is because that the advertiser does not enter the stage 2 during the bidding of these impression opportunities in the RAS. In the RL training process, the advertiser at each time step $t$ bids $a_t$ based on the current state $s_t$, and wins the impression opportunity $j$ with value $v_{j,t}$ if $a_t v_{j,t}^2 > p_{j,t}^2$, or loses otherwise. Once winning the impression opportunity $j$, the advertiser earns the value $v_{j,t}$ as the and pays the market price $p_{j,t}$. Note that the influence of all other advertisers are encoded in the market price $p_{j,t}^2$. However, the values $v_{j,t}^2$, $v_{j,t}$ and market prices $p_{j,t}^2$, $p_{j,t}$ are all constants during the RL training process.

## A.2   Simulated Advertising System Experiments

We construct a simulated advertising system s-RAS and build a simulated virtual advertising system s-VAS based on it. Specifically, the s-RAS is composed of two consecutive stages, where the auction mechanisms resemble those in the RAS. We consider the bidding process in a day, where the episode is divided into 96 time steps. Thus, the duration between any two adjacent time steps $t$ and $t+1$ is 15 minutes. The number of impression opportunities between time step $t$ and $t+1$ fluctuates from 100 to 500. Detailed parameters in the s-RAS is shown in Table. 1.

Table 1: The parameters used in the s-RAS.

| Parameters | Values |
| --- | --- |
| Number of advertisers | 100 |
| Time steps in an episode, $T$ | 96 |
| Minimum number of impression opportunities $N_t$ | 100 |
| Maximum number of impression opportunities $N_t$ | 500 |
| Minimum budget | $31,000$ Yuan |
| Maximum budget | $36,000$ Yuan |
| Value of impression opportunities in stage 1, $v_{j,t}^1$ | $0 \sim 1$ |
| Value of impression opportunities in stage 2, $v_{j,t}^2$ | $0 \sim 1$ |
| Minimum bidding price, $A_{\min}$ | $0$ Yuan |
| Maximum bidding price, $A_{\max}$ | $1,000$ Yuan |
| Maximum value of impression opportunity, $v_M$ | $1$ |
| Maximum market price, $p_M$ | $1,000$ Yuan |

We adopt the standard RL algorithm, DDPG [16], to train the auto-bidding policy of an advertiser in the s-RAS while keeping the policies of all other 99 advertisers fixed. Obviously, all other advertisers are viewed as parts of the environment with respect to the training advertiser. Fixing other advertisers' policies makes the environment stationary. The hyper-parameters used in the DDPG are shown in Table. 2. In addition, the s-VAS is built based on the historical data of the advertiser when bidding in stage 2 of the s-RAS, including the indexes of impression opportunities and the corresponding values and market prices. Moreover, an offline dataset is construct by collecting data directly from the s-RAS. We train the auto-bidding policy with the s-VAS and the offline dataset using the DDPG, where the hyper-parameters are the same as those in Table. 2. The differences in impression opportunities and

market prices between the s-RAS and s-VAS are shown in Fig. 2(a) in the manuscript and Fig. 2(b) in the manuscript, respectively. The RL rewards of these three settings are shown in Fig. 2(c) in the manuscript.

Table 2: The hyper-parameters of DDPG when training with the s-RAS, s-VAS and the offline dataset.

| Hyper-parameters | Values |
|---|---|
| Optimizer | Adam |
| Learning rate for critic network | $1 \times 10^{-4}$ |
| Learning rate for actor network | $1 \times 10^{-4}$ |
| Soft updated rate | 0.01 |
| Buffer size | 1000 |
| Sampling size | 200 |
| Discounted factor $\gamma$ | 0.99 |
| Random seeds | $1 \sim 16$ |
| Exploration actions | Gaussian noise with variance 0.01 |

## A.3 Illustrations of IBOO

Fig. A.3 illustrates the dominated gaps and influence of the IBOO.

## A.4 Importance and Universality of IBOO Problem

As stated in the manuscript, it was previously believed that training auto-bidding policies directly in the RAS is nearly impossible due to safety concerns. State-of-the-art auto-bidding policies training in the VAS face IBOO problems which can largely degrade their performance in the RAS. Solving the IBOO problem in the auto-bidding acts as the motivation of our work. Here, we further discuss the importance and universality of this motivation.

**Importance.** Online advertising business has become one of the main profit models for many companies, such as Google, Amazon, Alibaba, etc. In 2021, Google's online advertising revenue accounts for 82% of the total, and online advertising revenue in Alibaba accounts for over 90% of the total. At the same time, online advertising business also offers clients, acting as advertisers, a good chance to increase the ROI. Hence, online advertising business plays an important role for both companies and advertisers in the era of Internet. Recently, auto-bidding technique has become one of the most important tools for advertisers to lift up their ROI. However, state-of-the-art auto-bidding policies leveraging RL algorithms suffer from the IBOO problem. The IBOO can be significant when the gap between the optimal auto-bidding policy and the auto-bidding policy used for collecting data to construct the VAS is large. This is because many impression opportunities and corresponding information on values and market prices in the *optimal VAS*[2] are missing in the VAS. Hence, the auto-bidding policy cannot know how to behave on these unseen impression opportunities, and the improvement of the auto-bidding policy can be limited.

**Universality.** The IBOO problem does not exist only in the realm of auto-bidding. Actually, it exists in many other fields such as robotics [9], thermal power generating [25], and even computer visions [8], where the real-world environment cannot be accessed during RL training process and a virtual environment is needed. In these fields, the IBOO problem is usually known as the *sim2real* problem. Although many algorithms have been proposed to mitigate the IBOO (or sim2real) problem, it still remains a major challenge in the RL applications.

---

[2]For convenience, we name the VAS built based on the data collected by the optimal auto-bidding policy as the optimal VAS.

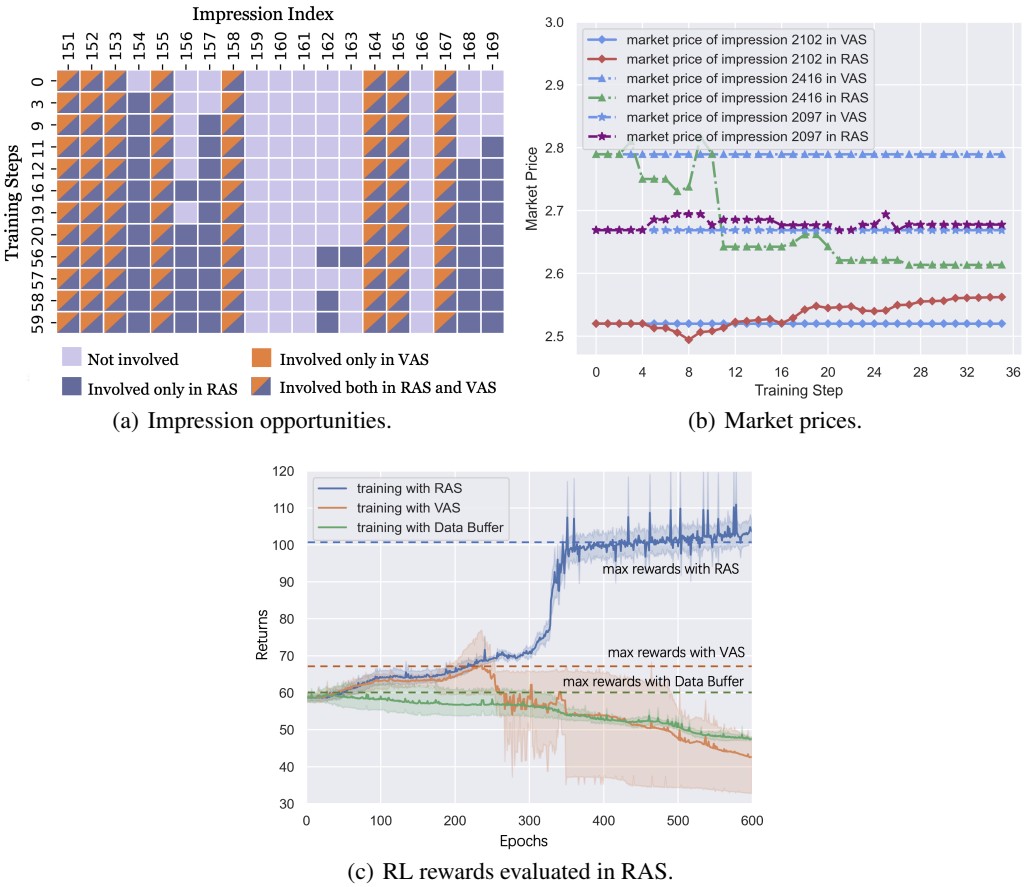

(a) Impression opportunities.

(b) Market prices.

(c) RL rewards evaluated in RAS.

Figure 2: Dominated gaps and IBOO influence. (a) shows that the impression opportunities within a certain period involving in the bidding of an advertiser remains the same in the VAS, but changes in stage 2 of the RAS during RL training process. (b) shows that the market prices remain the same in the VAS, but can rise, decrease or fluctuate around during RL training. (c) shows that auto-bidding policies training with the RAS can achieve higher rewards than those training with the VAS and a fixed data buffer using traditional RL methods.

# B   Related Work

In addition, to avoid IBOO, one may consider training auto-bidding policies with traditional RL algorithms based on the data collected by some safe policies directly from the RAS. However, this approach will suffer from extrapolation errors that can seriously degrade the policy's performance in RAS [11, 27]. As shown in Fig. 2(c), the expected cumulative rewards of traditional RL method [16] training with a fixed data buffer are lower than those of traditional RL method training with RAS and VAS. Though we can leverage offline RL techniques [11, 13, 30, 31] to mitigate this challenge, we cannot guarantee that the collected data contains sufficient transitions from high-reward regions [27]. This will strain the capacity of the offline RL algorithms to train near-optimal auto-bidding policies. Thus, extra data collections with different behavior policies (presumably better behavior policies) from the RAS are required. The policies trained by the offline RL methods cannot be directly used as behavior policies for data collections in the RAS, since the performance variance of the trained policies can be large [24]. Off policy evaluations (OPEs) have been recently studied for selecting policies with good performance without applying them to real-world environments [12, 32]. However, existing OPEs in auto-bidding are usually conducted in the VAS and can be inaccurate (see the last paragraph in this section). Therefore, we may still have little confidence to widely apply the policies trained by offline RL methods to advertisers for further online explorations in the RAS, even though they are proven to perform well by OPE. Note that there exist some safe online RL methods for safely

exploring in the environments [17, 20, 22, 21, 19, 18]. However, they are either developed based on the constraints that are not suitable for the auto-bidding problem or designed for systems with specific assumptions. Recently, with the development of offline RL methods, many algorithms for efficient online explorations on the premise of having an offline dataset [14, 23] have emerged. However, they often focus on the efficiency of RL training process rather than the safety of explorations.

**Safe Online RL.** [34] realize safe explorations by adding a safety layer at the end of the actor network. However, the safety layer needs to be trained by a prior dataset and can be inaccurate at states outside the dataset. [35] uses offline safety tests to examine the safety of the latest policy and directly applies it to explore online if it passes the tests. However, as we stated in Appendix G.2, there is no such reliable offline safety tests in auto-bidding. Besides, [21] realizes safe explorations by gradually increasing the attraction regions of the initial safe policy. However, it leverages the assumption that the environment is a linear model, which is not suitable for auto-bidding. As for this paper, based on the proved Lipschitz property of Q functions, we design the exploration policy by offsetting the actions to the promising directions relative to an initial safe policy.

**Extrapolation Error.** Extrapolation error means the misestimation of the states and actions outside the fixed dataset. A typical misestimation happens to the Q function in the standard RL algorithm. The fixed dataset cannot contain all the data from the environment, since the amount of all the data can usually be infinite. Hence, the trained Q function can only be accurate at the states and actions inside the dataset and can be inaccurate (usually overestimated) at those outside the dataset. This will make the actor network learn actions that extremely deviate from the behavioral actions and often bias towards bad actions. Hence, the policy performance can be seriously degraded. Offline RL algorithms usually address this challenge in three ways, including *policy constraint* methods, where explicit or implicit constraints are directly imposed to policies, such as BCQ [11], BEAR [30], and *conservative regularization* methods, where penalties for out-of-distribution (OOD) actions are imposed to the Q function, such as CQL [13], BRAC [40], as well as *modifications of imitation learning* method [37] such as ABM [31], CRR [38], BAIL[39].

**OPE in Auto-bidding.** Generally, the OPE used in auto-bidding is evaluate the auto-bidding policies in a VAS which is built based on the historical data of hundreds of advertisers. In the VAS, as we can know all the impression opportunities as well as their values and market prices in advance, we can calculate the optimal bids using linear programming [5]. Hence, the optimal accumulated rewards can be obtained. We define the ratio between the accumulated reward of the evaluated policy and the optimal accumulated rewards as $R/R^*$, which acts an important metric in the OPE of auto-bidding. The range of $R/R^*$ is $[0, 1]$. The closer the value of $R/R^*$ to 1, the better the performance of the evaluated auto-bidding policy. However, due to the IBOO, this common OPE method is not very accurate. Specifically, a low value of $R/R^*$ (below 0.7) can indicate a poor performance of the evaluated auto-bidding policy, while a large value of $R/R^*$ (above 0.8) does not indicate that the evaluated auto-bidding can certainly perform well in the RAS. Nonetheless, auto-bidding policies with higher $R/R^*$ are more likely to perform well in the RAS than those with lower $R/R^*$.

# C  Rationality Analysis of Assumptions

## C.1  Rationality of Assumption 1

**Assumption 1** (Bounded Impression Distributions). *Between time step $t$ and $t + 1$, we assume the numbers of winning impressions with action $a_t$ in the first stage $n_{t,1}$ and the second stage $n_{t,2}$ can both be bounded by linear functions, i.e., $n_{t,1} \leq k_1 a_t, n_{t,2} \leq k_2 a_t$, where $k_1, k_2 > 0$ are constants.*

In a stable RAS, the amount of increased (or decreased) winning impression opportunities for an advertiser when increasing (or reducing) the bids $a_t$ within any time step $t$ and $t + 1$ in both stage 1 and 2 will not change dramatically. Otherwise advertisers can largely increase the number of winning opportunities by slightly raising the bids, which can make the RAS unstable. Hence, there exist linear functions that can bound the changes in the amount of winning impression opportunities in both stage 1 and 2, where the slopes $k_1$ and $k_2$ have limited values (that usually are not very large). Fig. 3 illustrates this assumption with the data generated in an bidding episode in the s-RAS.

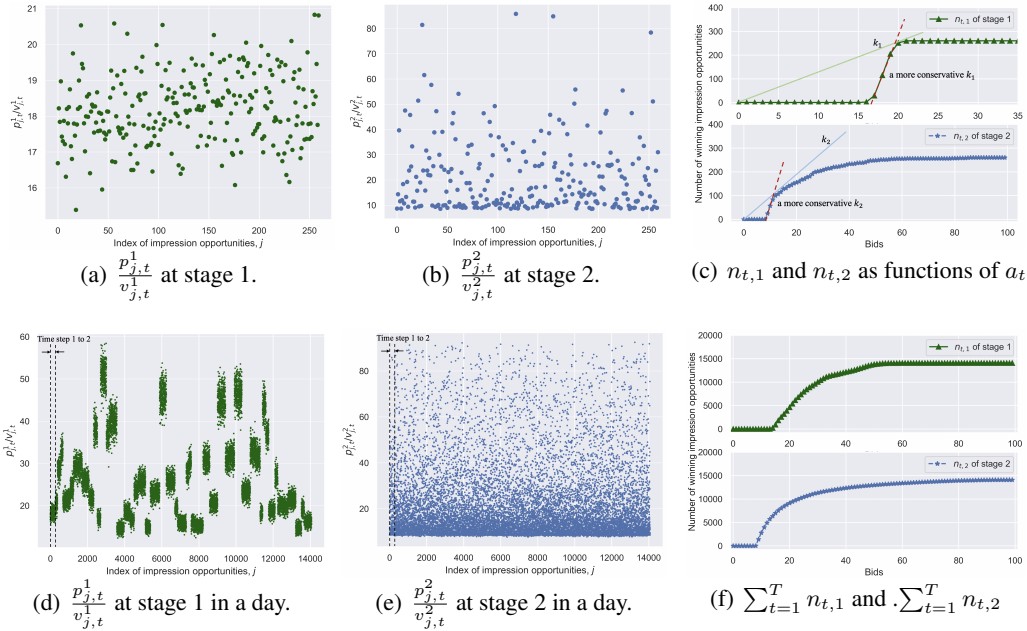

(a) $\frac{p_{j,t}^1}{v_{j,t}^1}$ at stage 1.

(b) $\frac{p_{j,t}^2}{v_{j,t}^2}$ at stage 2.

(c) $n_{t,1}$ and $n_{t,2}$ as functions of $a_t$

(d) $\frac{p_{j,t}^1}{v_{j,t}^1}$ at stage 1 in a day.

(e) $\frac{p_{j,t}^2}{v_{j,t}^2}$ at stage 2 in a day.

(f) $\sum_{t=1}^{T} n_{t,1}$ and .$\sum_{t=1}^{T} n_{t,2}$

Figure 3: The fractions between the market price and value of impression opportunities as well as the number of winning impression opportunities changing with the bids $a_t$. (a) to (c) present the data between time step 1 and 2, and (d) to (e) displays the data of the whole episode (a day).

## C.2    Rationality of Assumption 2

**Assumption 2** (Bounded Partial Derivations of $Q^\mu(s_t, a_t)$). *We assume that the partial derivation of $Q^\mu(s_t, a_t)$ with respect to $s_t(1)$ and $s_t(3)$ is bounded, i.e., $\left|\frac{\partial Q^\mu(s_t, a_t)}{\partial s_t(1)}\right| \leq k_3$ and $\left|\frac{\partial Q^\mu(s_t, a_t)}{\partial s_t(3)}\right| \leq k_4$, where $k_3, k_4 > 0$ are constants.*

Recall that the value of the Q function at state $s_t$ and action $a_t$ represents the total value of winning impression opportunities starting from state $s_t$, bidding with $a_t$ and following policy $\mu$ afterwards. In a stable RAS, this total value of winning impression opportunities will not increase (or drop) dramatically when the advertiser slightly increases (or reduces) the budget. Similarly, $Q^\mu(s_t, a_t)$ will not extremely change if the advertiser spends a little more budget before time step $t$. Hence, the absolute values of the partial derivatives of $Q^\mu(s_t, a_t)$ with respect to the budget left $s_t(1)$ and the consumed budget $s_t(2)$ have limited values (that are usually not very large). We denote the upper bounds of $\left|\frac{\partial Q^\mu(s_t, a_t)}{\partial s_t(1)}\right|$ and $\left|\frac{\partial Q^\mu(s_t, a_t)}{\partial s_t(3)}\right|$ as $k_3$ and $k_4$, respectively.

# D    Proofs of Propositions

## D.1    Proofs of Proposition 1

**Proposition 1** (Analytical expressions of $R$, $C$ and $\mathbb{P}$). *Based on the characteristic of the two-stage cascaded auction in the RAS, we can formulate the reward function $R$ as $r_t(s_t, a_t) = \sum_j \mathbb{1}\{a_t v_{j,t}^1 \geq p_{j,t}^1, a_t v_{j,t}^2 \geq p_{j,t}^2\} v_{j,t}$, the constraint function $C$ as $c_t(s_t, a_t) = \sum_j \mathbb{1}\{a_t v_{j,t}^1 \geq p_{j,t}^1, a_t v_{j,t}^2 \geq p_{j,t}^2\} p_{j,t}$, and the state transition rule $\mathbb{P}$ as $s_{t+1} = s_t + [\triangle s_t(1), \triangle s_t(2), \triangle s_t(3)]$, where $\triangle s_t(1) = -\triangle s_t(3) = -\sum_j \mathbb{1}\{a_t v_{j,t}^1 \geq p_{j,t}^1, a_t v_{j,t}^2 \geq p_{j,t}^2\} p_{j,t}$ and $\triangle s_t(2) = -1$. Note that $p_{j,t}^1$ and $v_{j,t}^1$ denote the market price and rough value of impression $j$ in stage 1, and $p_{j,t}^2$ and $v_{j,t}^2$ denote the market price and accurate value of impression $j$ in stage 2.*

*Proof.* As stated in Appendix A.1, the auction is completed in two cascaded stages. The condition to win the impression opportunity $j$ for an advertiser is that

- successfully passing the stage 1, i.e., $a_t v_{j,t}^1 \geq p_{j,t}^1$, and

- bidding the highest in stage 2, i.e., $a_t v_{j,t}^2 \geq p_{j,t}^2$

hold at the same time. Hence, the reward function which is the total value of winning impression opportunities between time step $t$ and $t+1$ can be expressed as

$$r_t(s_t, a_t) = \sum_j \mathbb{1}\left\{ a_t v_{j,t}^1 \geq p_{j,t}^1, a_t v_{j,t}^2 \geq p_{j,t}^2 \right\} v_{j,t}, \tag{2}$$

and the cost function can be expressed as

$$c_t(s_t, a_t) = \sum_j \mathbb{1}\left\{ a_t v_{j,t}^1 \geq p_{j,t}^1, a_t v_{j,t}^2 \geq p_{j,t}^2 \right\} p_{j,t}. \tag{3}$$

The amount of the consumed budget between time step $t$ and $t+1$ can be expressed as

$$b_t - b_{t+1} = -\triangle s_t(1) = \triangle s_t(3) = \sum_j \mathbb{1}\{a_t v_{j,t}^1 \geq p_{j,t}^1, a_t v_{j,t}^2 \geq p_{j,t}^2\} p_{j,t}. \tag{4}$$

Besides, $\triangle s_t(2) = T - (t+1) - T + t = -1$. $\qquad\square$

# E    Proofs of Theorems

## E.1    Proof of Theorem 1

**Theorem 1** (Lipschitz Smooth of $r_t(s_t, a_t)$)**.** *Under Assumption 1, the reward function $r_t(s_t, a_t)$ is $L_r$-Lipschitz smooth with respect to actions $a_t$ at any given state $s_t$, where $L_r = (k_1 + k_2)v_M$.*

*Proof.* Recall that the reward function $r_t(s_t, a_t)$ can be expressed as

$$r_t(s_t, a_t) = \sum_{j=1}^{N_t} \mathbb{1}\{a_t v_{j,t}^1 \geq p_{j,t}^1, a_t v_{j,t}^2 \geq p_{j,t}^2\} v_{j,t} = \sum_{j=1}^{N_t} \mathbb{1}\left\{ a_t \geq \frac{p_{j,t}^1}{v_{j,t}^1}, a_t \geq \frac{p_{j,t}^2}{v_{j,t}^2} \right\} v_{j,t}. \tag{5}$$

Hence, $\forall s_t \in \mathcal{S}$ and $\forall a_1, a_2 \in \mathcal{A}, a_1 \neq a_2$, we have

$$|r_t(s_t, a_1) - r_t(s_t, a_2)| = \left| \sum_{j=1}^{N_t} \left[ \mathbb{1}\left\{ a_1 \geq \frac{p_{j,t}^1}{v_{j,t}^1}, a_1 \geq \frac{p_{j,t}^2}{v_{j,t}^2} \right\} - \mathbb{1}\left\{ a_2 \geq \frac{p_{j,t}^1}{v_{j,t}^1}, a_2 \geq \frac{p_{j,t}^2}{v_{j,t}^2} \right\} \right] v_{j,t} \right|. \tag{6}$$

Without loss of generality, we let $a_1 > a_2$. Note that the advertiser can win any impression opportunity $j$ with bid price $a_1$ if it can win this impression opportunity with bid price $a_2$, which means

$$\mathbb{1}\left\{ a_1 \geq \frac{p_{j,t}^1}{v_{j,t}^1}, a_1 \geq \frac{p_{j,t}^2}{v_{j,t}^2} \right\} \geq \mathbb{1}\left\{ a_2 \geq \frac{p_{j,t}^1}{v_{j,t}^1}, a_2 \geq \frac{p_{j,t}^2}{v_{j,t}^2} \right\}. \tag{7}$$

Thus, we can drop the absolute value sign in (6) and obtain

$$|r_t(s_t, a_1) - r_t(s_t, a_2)| = \sum_{j=1}^{N_t} \left[ \mathbb{1}\left\{ a_1 \geq \frac{p_{j,t}^1}{v_{j,t}^1}, a_1 \geq \frac{p_{j,t}^2}{v_{j,t}^2} \right\} - \mathbb{1}\left\{ a_2 \geq \frac{p_{j,t}^1}{v_{j,t}^1}, a_2 \geq \frac{p_{j,t}^2}{v_{j,t}^2} \right\} \right] v_{j,t}$$

$$\leq v_M \sum_{j=1}^{N_t} \left[ \mathbb{1}\left\{ a_1 \geq \frac{p_{j,t}^1}{v_{j,t}^1}, a_1 \geq \frac{p_{j,t}^2}{v_{j,t}^2} \right\} - \mathbb{1}\left\{ a_2 \geq \frac{p_{j,t}^1}{v_{j,t}^1}, a_2 \geq \frac{p_{j,t}^2}{v_{j,t}^2} \right\} \right]$$

$$= v_M \sum_{j=1}^{N_t} \left[ \mathbb{1}\left\{ a_1 \geq \frac{p_{j,t}^1}{v_{j,t}^1}, a_1 \geq \frac{p_{j,t}^2}{v_{j,t}^2}, \left( a_2 < \frac{p_{j,t}^1}{v_{j,t}^1}, \text{or } \frac{p_{j,t}^1}{v_{j,t}^1} \leq a_2 < \frac{p_{j,t}^2}{v_{j,t}^2} \right) \right\} \right]$$

$$= v_M \sum_{j=1}^{N_t} \left[ \mathbb{1}\left\{ a_1 \geq \frac{p_{j,t}^1}{v_{j,t}^1}, a_1 \geq \frac{p_{j,t}^2}{v_{j,t}^2}, a_2 < \frac{p_{j,t}^1}{v_{j,t}^1} \right\} + \right.$$

$$\left. \mathbb{1}\left\{ a_1, a_2 \geq \frac{p_{j,t}^1}{v_{j,t}^1}, a_1 \geq \frac{p_{j,t}^2}{v_{j,t}^2}, a_2 < \frac{p_{j,t}^2}{v_{j,t}^2} \right\} \right] \tag{8}$$

Note that (8) use the fact that the additional impression opportunities won by bid $a_1$ compared to bid $a_2$, i.e., $\sum_{j=1}^{N_t} \left[ \mathbb{1}\left\{ a_1 \geq \frac{p_{j,t}^1}{v_{j,t}^1}, a_1 \geq \frac{p_{j,t}^2}{v_{j,t}^2} \right\} - \mathbb{1}\left\{ a_2 \geq \frac{p_{j,t}^1}{v_{j,t}^1}, a_2 \geq \frac{p_{j,t}^2}{v_{j,t}^2} \right\} \right]$, can be divided into two parts:

- the first part are the impression opportunities that can be won with bid $a_1$ but cannot be won with bid $a_2$ even in stage 1, i.e., $\sum_{j=1}^{N_t} \mathbb{1}\left\{ a_1 \geq \frac{p_{j,t}^1}{v_{j,t}^1}, a_1 \geq \frac{p_{j,t}^2}{v_{j,t}^2}, a_2 < \frac{p_{j,t}^1}{v_{j,t}^1} \right\}$;

- the second part are the impression opportunities that can be won in the stage 1 with both bids $a_1$ and $a_2$, but can only be won in stage 2 with bid $a_1$, not $a_2$, i.e., $\sum_{j=1}^{N_t} \mathbb{1}\left\{ a_1, a_2 \geq \frac{p_{j,t}^1}{v_{j,t}^1}, a_1 \geq \frac{p_{j,t}^2}{v_{j,t}^2}, a_2 < \frac{p_{j,t}^2}{v_{j,t}^2} \right\}$.

To illustrates these two parts of impression opportunities, an example of bidding with $a_1$ and $a_2$ in stage 1 and stage 2 is shown in Fig. 4. The first part of impression opportunities can be bounded by:

$$\sum_{j=1}^{N_t} \mathbb{1}\left\{ a_1 \geq \frac{p_{j,t}^1}{v_{j,t}^1}, a_1 \geq \frac{p_{j,t}^2}{v_{j,t}^2}, a_2 < \frac{p_{j,t}^1}{v_{j,t}^1} \right\} \leq \sum_{j=1}^{N_t} \mathbb{1}\left\{ a_1 \geq \frac{p_{j,t}^1}{v_{j,t}^1} > a_2 \right\}, \tag{9}$$

which is represented by the red shaded area in Fig. 4(a). Similarly, the second part of impression opportunities can be bounded by:

$$\sum_{j=1}^{N_t} \mathbb{1}\left\{ a_1, a_2 \geq \frac{p_{j,t}^1}{v_{j,t}^1}, a_1 \geq \frac{p_{j,t}^2}{v_{j,t}^2}, a_2 < \frac{p_{j,t}^2}{v_{j,t}^2} \right\} \leq \sum_{j=1}^{N_t} \mathbb{1}\left\{ a_1 \geq \frac{p_{j,t}^2}{v_{j,t}^2} > a_2 \right\}, \tag{10}$$

which is represented by the red shaded area in Fig. 4(d). Hence, with Assumption 1, we have

$$\left| r_t(s_t, a_1) - r_t(s_t, a_2) \right| \leq v_M \sum_{j=1}^{N_t} \left[ \mathbb{1}\left\{ a_1 \geq \frac{p_{j,t}^1}{v_{j,t}^1} > a_2 \right\} + \mathbb{1}\left\{ a_1 \geq \frac{p_{j,t}^2}{v_{j,t}^2} > a_2 \right\} \right]$$

$$\leq (k_1 + k_2) v_M |a_1 - a_2|. \tag{11}$$

The upper bound of the changing rate of the reward function $r_t(s_t, a_t)$ is

$$\frac{\left| r_t(s_t, a_1) - r_t(s_t, a_2) \right|}{|a_1 - a_2|} \leq (k_1 + k_2) v_M, \tag{12}$$

which indicates that $r_t(s_t, a_t)$ is $L_r$-Lipschitz smooth, $L_r \triangleq (k_1 + k_2) v_M$. $\qquad \square$

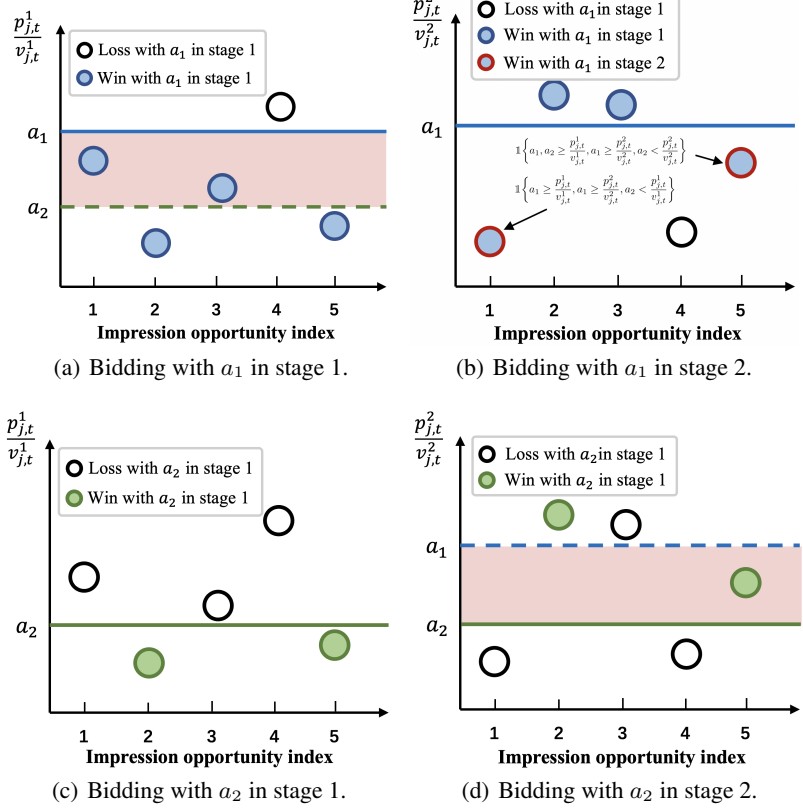

(a) Bidding with $a_1$ in stage 1.

(b) Bidding with $a_1$ in stage 2.

(c) Bidding with $a_2$ in stage 1.

(d) Bidding with $a_2$ in stage 2.

Figure 4: Bidding with $a_1$ and $a_2$ in stage 1 and stage 2, where $a_1 > a_2$. The extra impression opportunities won by bid $a_1$ compared to bid $a_2$ are impression opportunity 1 that satisfies $a_1 \geq \frac{p_{1,t}^1}{v_{1,t}^1}, a_1 \geq \frac{p_{1,t}^2}{v_{1,t}^2}, a_2 < \frac{p_{1,t}^2}{v_{1,t}^2}$, and impression 5 that satisfies $a_1, a_2 \geq \frac{p_{5,t}^1}{v_{5,t}^1}, a_1 \geq \frac{p_{5,t}^2}{v_{5,t}^2} \geq a_2$.

## E.2 Proof of Theorem 2

**Theorem 2** (Lipschitz Smooth of $Q^\mu(s_t, a_t)$). *Under Assumption 1 and 2, the Q function $Q^\mu(s_t, a_t)$ is an $L_Q$-Lipschitz smooth function with respect to the actions $a_t$ at any given state $s_t$, where $L_Q = [v_M + (k_3 + k_4)p_M](k_1 + k_2)$.*

*Proof.* Recall that $Q^\mu(s_t, a_t)$ can be expressed as

$$Q^\mu(s_t, a_t) = r_t(s_t, a_t) + \gamma \mathbb{E}_{s_{t+1} \sim \mathbb{P}(\cdot|s_t, a_t)} Q^\mu(s_{t+1}, \mu(s_{t+1})), \tag{13}$$

where $r_t(s_t, a_t)$ is Lipschitz smooth. Thus, we first focus on the characteristic of the second part $\gamma \mathbb{E}_{s_{t+1} \sim \mathbb{P}(\cdot|s_t, a_t)} Q^\mu(s_{t+1}, \mu(s_{t+1}))$. According to Proposition 1, at any given state $s_t \in \mathcal{S}$, the next states $s_{t+1}^1$ and $s_{t+1}^2$ under bids $a_1$ and $a_2$ can be expressed as

$$s_{t+1}^1 = s_t + [\triangle s_1(1), \triangle s_1(2), \triangle s_1(3)], \quad s_{t+1}^2 = s_t + [\triangle s_2(1), \triangle s_2(2), \triangle s_2(3)], \tag{14}$$

where

$$\triangle s_1(1) = -\triangle s_1(3) = -\sum_{j=1}^{N_t} \mathbb{1}\left\{a_1 \geq \frac{p_{j,t}^1}{v_{j,t}^1}, a_1 \geq \frac{p_{j,t}^2}{v_{j,t}^2}\right\} p_{j,t}, \tag{15}$$

and

$$\triangle s_2(1) = -\triangle s_2(3) = -\sum_{j=1}^{N_t} \mathbb{1}\left\{a_2 \geq \frac{p_{j,t}^1}{v_{j,t}^1}, a_2 \geq \frac{p_{j,t}^2}{v_{j,t}^2}\right\} p_{j,t}. \tag{16}$$

and

$$\triangle s_1(2) = \triangle s_2(2) = -1. \tag{17}$$

Hence, using Taylor expansion, we have

$$\left| \mathbb{E}_{s_{t+1}^1 \sim \mathbb{P}(\cdot|s_t,a_1)} Q^\mu(s_{t+1}^1, \mu(s_{t+1}^1)) - \mathbb{E}_{s_{t+1}^2 \sim \mathbb{P}(\cdot|s_t,a_2)} Q^\mu(s_{t+1}^2, \mu(s_{t+1}^2)) \right|$$

$$= \left| \mathbb{E}_{s_{t+1}^1 \sim \mathbb{P}(\cdot|s_t,a_1)} Q^\mu(s_{t+1}^1) - \mathbb{E}_{s_{t+1}^2 \sim \mathbb{P}(\cdot|s_t,a_2)} Q^\mu(s_{t+1}^2) \right|$$

$$= \left| Q^\mu(s_t + [\triangle s_1(1), \triangle s_1(2), \triangle s_1(3)]) - Q^\mu(s_t + [\triangle s_2(1), \triangle s_2(2), \triangle s_2(3)]) \right|$$

$$\approx \left| Q^\mu(s_t) + \frac{\partial Q^\mu(s_t)}{\partial s_t(1)}\triangle s_1(1) + \frac{\partial Q^\mu(s_t)}{\partial s_t(2)}\triangle s_1(2) + \frac{\partial Q^\mu(s_t)}{\partial s_t(3)}\triangle s_1(3) - Q^\mu(s_t) - \frac{\partial Q^\mu(s_t)}{\partial s_t(1)}\triangle s_2(1) \right.$$
$$\left. - \frac{\partial Q^\mu(s_t)}{\partial s_t(2)}\triangle s_2(2) - \frac{\partial Q^\mu(s_t)}{\partial s_t(3)}\triangle s_2(3) \right|$$

$$= \left| \frac{\partial Q^\mu(s_t)}{\partial s_t(1)}\triangle s_1(1) + \frac{\partial Q^\mu(s_t)}{\partial s_t(3)}\triangle s_1(3) - \frac{\partial Q^\mu(s_t)}{\partial s_t(1)}\triangle s_2(1) - \frac{\partial Q^\mu(s_t)}{\partial s_t(3)}\triangle s_2(3) \right|$$

$$= \left| \left( \frac{\partial Q^\mu(s_t)}{\partial s_t(1)} - \frac{\partial Q^\mu(s_t)}{\partial s_t(3)} \right) \left( \sum_{j=1}^{N_t} \left[ -\mathbb{1}\left\{ a_1 \geq \frac{p_{j,t}^1}{v_{j,t}^1}, \frac{p_{j,t}^2}{v_{j,t}^2} \right\} + \mathbb{1}\left\{ a_2 \geq \frac{p_{j,t}^1}{v_{j,t}^1}, \frac{p_{j,t}^2}{v_{j,t}^2} \right\} \right] p_{j,t}. \right) \right|$$

$$\leq \left| \left( \frac{\partial Q^\mu(s_t)}{\partial s_t(1)} - \frac{\partial Q^\mu(s_t)}{\partial s_t(3)} \right) \right| \left| -r_t(s_t,a_1) + r_t(s_t,a_2) \right| \frac{p_M}{v_M}$$

$$\leq \left( \left| \frac{\partial Q^\mu(s_t)}{\partial s_t(1)} \right| + \left| \frac{\partial Q^\mu(s_t)}{\partial s_t(3)} \right| \right) (k_1 + k_2) p_M |a_1 - a_2|$$

$$= (k_1 + k_2)(k_3 + k_4) p_M |a_1 - a_2|. \tag{18}$$

Note that we use (11) in Theorem 1. Hence, we have

$$\left| Q^\mu(s_t,a_1) - Q^\mu(s_t,a_2) \right| \leq \left| r_t(s_t,a_1) - r_t(s_t,a_2) \right| +$$

$$\gamma \left| \mathbb{E}_{s_{t+1}^1 \sim \mathbb{P}(\cdot|s_t,a_1)} Q^\mu(s_{t+1}^1) - \mathbb{E}_{s_{t+1}^2 \sim \mathbb{P}(\cdot|s_t,a_2)} Q^\mu(s_{t+1}^2) \right|$$

$$\leq (k_1 + k_2) v_M \left| a_1 - a_2 \right| + \gamma(k_1 + k_2)(k_3 + k_4) p_M \left| a_1 - a_2 \right|$$

$$= \left[ v_M + \gamma(k_3 + k_4) p_M \right] (k_1 + k_2) \left| a_1 - a_2 \right|. \tag{19}$$

The upper bound of the absolute changing rate of the Q function $Q^\mu(s_t,a_t)$ is

$$\frac{\left| Q^\mu(s_t,a_1) - Q^\mu(s_t,a_2) \right|}{\left| a_1 - a_2 \right|} \leq \left[ v_M + \gamma(k_3 + k_4) p_M \right] (k_1 + k_2), \tag{20}$$

which indicates that $Q^\mu(s_t,a_t)$ is $L_Q$-Lipschitz smooth, $L_Q \triangleq [v_M + \gamma(k_3 + k_4)p_M](k_1 + k_2)$.  □

## E.3  Proof of Theorem 3

**Theorem 3** ( Upper Bound of $|V(\pi) - V(\mu_s)|$)**.** *The expected accumulated reward $V(\pi_e)$ satisfies*

$$\left| V(\pi_e) - V(\mu_s) \right| \leq \xi \gamma^{t_1} \left[ v_M + \gamma(k_3 + k_4) p_M \right] (k_1 + k_2) \Delta T. \tag{21}$$

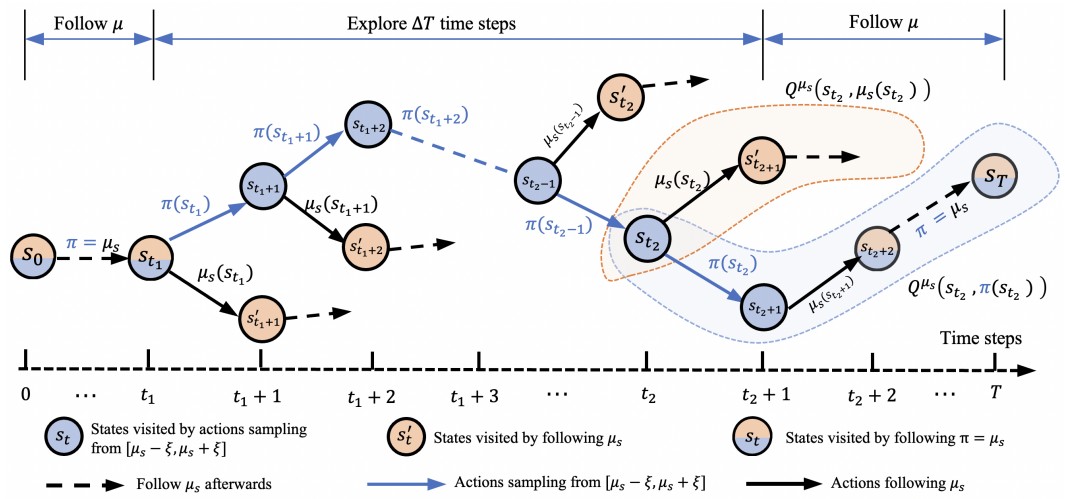

Figure 5: Practical exploration policy $\pi$ based on safe policy $\mu_s$.

*Proof.* Fig. 5 shows the visited states during one episode using exploration policy $\pi$. The total value $V(\pi)$ can be expressed as [3]:

$$
\begin{aligned}
V(\pi) &= \mathbb{E}_{a_t \sim \pi}\left[\sum_{t=0}^{T-1} \gamma^t r_t(s_t, a_t) \,\middle|\, s_0 \sim \rho_0\right] \\
&= \mathbb{E}_{a_t \sim \mu_s}\left[\sum_{t=0}^{t_1-1} \gamma^t r_t(s_t, a_t) \,\middle|\, s_0 \sim \rho_0\right] + \mathbb{E}_{a_t \sim \pi}\left[\sum_{t=t_1}^{T-1} \gamma^t r_t(s_t, a_t) \,\middle|\, s_{t_1} \sim \rho_{t_1}\right],
\end{aligned}
\tag{22}
$$

where $\rho_t$ denotes the state distribution at time step $t$ starting from $s_0 \sim \rho_0$ and following $\pi$. The total value $V(\mu_s)$ is

$$
V(\mu_s) = \mathbb{E}_{a_t \sim \mu_s}\left[\sum_{t=0}^{T-1} \gamma^t r_t(s_t, a_t) \,\middle|\, s_0 \sim \rho_0\right].
\tag{23}
$$

Notice that the accumulated rewards from time step 0 to time step $t_1$ in both $V(\pi)$ and $V(\mu_s)$ are the same. Hence, the difference between $V(\pi)$ and $V(\mu_s)$ can be calculated as

$$
V(\pi) - V(\mu_s) = \underbrace{\mathbb{E}_{a_t \sim \pi}\left[\sum_{t=t_1}^{T-1} \gamma^t r_t(s_t, a_t) \,\middle|\, s_{t_1} \sim \rho_{t_1}\right]}_{①} - \mathbb{E}_{a_t \sim \mu_s}\left[\sum_{t=t_1}^{T-1} \gamma^t r_t(s_t, a_t) \,\middle|\, s_{t_1} \sim \rho_{t_1}\right].
$$

$$
\tag{24}
$$

The term ① can be further divided into three parts, including the accumulated rewards from time step $t$ to $t + \Delta T - 1$ (part 1), the immediate reward at time step $t + \Delta T$ (part 2) and the accumulated rewards from time step $t + \Delta T + 1$ to $T$ (part 3), i.e.,

---

[3]Note that the state transitions in all formulas follow the rule of $\mathbb{P}$, i.e., $s_{t+1} \sim \mathbb{P}(\cdot | s_t, a_t), \forall \tau \in \{0, 1, ..., T-1\}$. Hence, for brevity, we omit this term in the subscript of the expectation operator $\mathbb{E}$ in the following formulas.

$$\textcircled{1} = \mathbb{E}_{a_t \sim \pi}\left[\sum_{t=t_1}^{T-1} \gamma^t r_t(s_t, a_t)\Big| s_{t_1} \sim \rho_{t_1}\right] = \underbrace{\mathbb{E}_{a_t \sim \pi}\left[\sum_{t=t_1}^{t_2-1} \gamma^t r_t(s_t, a_t)\Big| s_{t_1} \sim \rho_{t_1}\right]}_{\text{part 1: accumulated rewards from } t_1 \text{ to } t_2 \text{ following } \pi}$$

$$+ \gamma^{t_2}\mathbb{E}_{s_{t_2} \sim \rho_{t_2}}\left[\underbrace{r_{t_2}(s_{t_2}, \pi(s_{t_2}))}_{\text{part 2: immediate reward at time step } t_2} + \gamma \underbrace{\mathbb{E}_{a_t \sim \mu_s, \forall t \geq t_2+1}\left[\sum_{t=t_2+1}^{T-1} \gamma^{t-t_2-1} r_t(s_t, a_t)\Big| s_{t_2+1} \sim \rho_{t_2+1}\right]}_{\text{part 3: accumulated rewards from } t_2 + 1 \text{ to } T - 1 \text{ following } \mu_s}\right]$$

$$\underbrace{\hphantom{+ \gamma^{t_2}\mathbb{E}_{s_{t_2} \sim \rho_{t_2}}\left[r_{t_2}(s_{t_2}, \pi(s_{t_2}))\right]}}_{\text{part 2+part 3} = Q^{\mu_s}(s_{t_2}, \pi(s_{t_2}))}$$

$$= \mathbb{E}_{a_t \sim \pi}\left[\sum_{t=t_1}^{t_2-1} \gamma^t r_t(s_t, a_t)\Big| s_{t_1} \sim \rho_{t_1}\right]$$

$$+ \gamma^{t_2}\mathbb{E}_{s_{t_2} \sim \rho_{t_2}}\left[Q^{\mu_s}(s_{t_2}, \pi(s_{t_2})) \underbrace{-Q^{\mu_s}(s_{t_2}, \mu_s(s_{t_2})) + Q^{\mu_s}(s_{t_2}, \mu_s(s_{t_2}))}_{\text{trick: plus and minus } Q^{\mu_s}(s_{t_2}, \mu_s(s_{t_2}))}\right]$$

$$= \underbrace{\mathbb{E}_{a_t \sim \pi}\left[\sum_{t=t_1}^{t_2-1} \gamma^t r_t(s_t, a_t)\Big| s_{t_1} \sim \rho_{t_1}\right]}_{\textcircled{2}} + \gamma^{t_2}\mathbb{E}_{s_{t_2} \sim \rho_{t_2}}\left[\underbrace{Q^{\mu_s}(s_{t_2}, \pi(s_{t_2})) - Q^{\mu_s}(s_{t_2}, \mu_s(s_{t_2}))}_{\triangleq \Delta Q(t_2)}\right]$$

$$+ \underbrace{\gamma^{t_2}\mathbb{E}_{s_{t_2} \sim \rho_{t_2}}\left[Q^{\mu_s}(s_{t_2}, \mu_s(s_{t_2}))\right]}_{\textcircled{3}}, \tag{25}$$

where we define $\Delta Q(t) \triangleq Q^{\mu_s}(s_t, \pi(s_t)) - Q^{\mu_s}(s_t, \mu_s(s_t))$. Note that we can take term $\gamma^{t_2-1} r_{t_2-1}(s_{t_2-1}, a_{t_2-1})$ from $\textcircled{2}$ and combine it with term $\textcircled{3}$ to obtain $Q^{\mu_s}(s_{t_2-1}, \pi(s_{t_2-1}))$. In fact, we have: $\forall \tau \in \{1, 2, ..., t_2 - t_1\}$,

$$\mathbb{E}_{a_t \sim \pi}\left[\sum_{t=t_1}^{t_2-\tau} \gamma^t r_t(s_t, a_t)\Big| s_{t_1} \sim \rho_{t_1}\right] + \gamma^{t_2-\tau+1}\mathbb{E}_{s_{t_2-\tau+1} \sim \rho_{t_2-\tau+1}}\left[Q^{\mu_s}(s_{t_2-\tau+1}, \mu_s(s_{t_2-\tau+1}))\right]$$

$$= \mathbb{E}_{a_t \sim \pi}\left[\sum_{t=t_1}^{t_2-\tau-1} \gamma^t r_t(s_t, a_t)\Big| s_{t_1} \sim \rho_{t_1}\right] + \gamma^{t_2-\tau}\left\{\mathbb{E}_{s_{t_2-\tau} \sim \rho_{t_2-\tau}}\left[r_{t_2-\tau}(s_{t_2-\tau}, \pi(s_{t_2-\tau}))\right]\right.$$

$$\left. + \gamma\mathbb{E}_{a_t \sim \mu_s, \forall t \geq t_2-\tau+1}\left[\sum_{t=t_2-\tau+1}^{T-1} \gamma^{t-t_2+\tau-1} r_t(s_t, a_t)\Big| s_{t_2-\tau+1} \sim \rho_{t_2-\tau+1}\right]\right\}$$

$$= \underbrace{\mathbb{E}_{a_t \sim \pi}\left[\sum_{t=t_1}^{t_2-\tau-1} \gamma^t r_t(s_t, a_t)\Big| s_{t_1} \sim \rho_{t_1}\right]}_{\textcircled{2}'} + \gamma^{t_2-\tau}\mathbb{E}_{s_{t_2-\tau} \sim \rho_{t_2-\tau}}\left[Q^{\mu_s}(s_{t_2-\tau}, \pi(s_{t_2-\tau}))\right]. \tag{26}$$

We can continue to use *plus and minus* trick as we did in (25) to further break down term ②'. Hence, term ① can be calculated as

$$
\begin{aligned}
① &= \mathbb{E}_{a_t \sim \pi}\left[\sum_{t=t_1}^{t_2-2}\gamma^t r_t(s_t,a_t)\Big|s_{t_1}\sim\rho_{t_1}\right] + \gamma^{t_2-1}\mathbb{E}_{s_{t_2-1}\sim\rho_{t_2-1}}\left[Q^{\mu_s}(s_{t_2-1},\mu_s(s_{t_2-1}))\right] \\
&\quad + \gamma^{t_2-1}\mathbb{E}_{s_{t_2-1}\sim\rho_{t_2-1}}\left[\Delta Q(t_2-1)\right] + \gamma^{t_2}\mathbb{E}_{s_{t_2}\sim\rho_{t_2}}\left[\Delta Q(t_2)\right] \\
&= \mathbb{E}_{a_t\sim\pi}\left[\sum_{t=t_1}^{t_2-3}\gamma^t r_t(s_t,a_t)\Big|s_{t_1}\sim\rho_{t_1}\right] + \gamma^{t_2-2}\mathbb{E}_{s_{t_2-2}\sim\rho_{t_2-2}}\left[Q^{\mu_s}(s_{t_2-2},\mu_s(s_{t_2-2}))\right] \\
&\quad + \gamma^{t_2-2}\mathbb{E}_{s_{t_2-2}\sim\rho_{t_2-2}}\left[\Delta Q(t_2-2)\right] + \gamma^{t_2-1}\mathbb{E}_{s_{t_2-1}\sim\rho_{t_2-1}}\left[\Delta Q(t_2-1)\right] + \gamma^{t_2}\mathbb{E}_{s_{t_2}\sim\rho_{t_2}}\left[\Delta Q(t_2)\right] \\
&= \cdots\cdots\cdots \\
&= \mathbb{E}_{a_t\sim\pi}\left[\gamma^{t_1}r_{t_1}(s_{t_1},a_{t_1})\Big|s_{t_1}\sim\rho_{t_1}\right] + \gamma^{t_1+1}\mathbb{E}_{s_{t_1+1}\sim\rho_{t_1+1}}\left[Q^{\mu_s}(s_{t_1+1},\mu_s(s_{t_1+1}))\right] \\
&\quad + \sum_{t=t_1+1}^{t_2}\gamma^t\mathbb{E}_{s_t\sim\rho_t}\left[\Delta Q(t)\right] \\
&= \gamma^{t_1}\mathbb{E}_{s_{t_1}\sim\rho_{t_1}}\left\{r_{t_1}(s_{t_1},\pi(s_{t_1})) + \gamma\mathbb{E}_{a_t\sim\mu_s,\forall t\geq t_1+1}\left[\sum_{t=t_1+1}^{T-1}\gamma^{t-t_1-1}r_t(s_t,a_t)\Big|s_{t_1+1}\sim\rho_{t_1+1}\right]\right\} \\
&\quad + \sum_{t=t_1+1}^{t_2}\gamma^t\mathbb{E}_{s_t\sim\rho_t}\left[\Delta Q(t)\right] \\
&= \gamma^{t_1}\mathbb{E}_{s_{t_1}\sim\rho_{t_1}}\left[Q^{\mu_s}(s_{t_1},\pi(s_{t_1}))\right] + \sum_{t=t_1+1}^{t_2}\gamma^t\mathbb{E}_{s_t\sim\rho_t}\left[\Delta Q(t)\right].
\end{aligned}
\tag{27}
$$

Substitute term ① in (24) by (27), we can obtain

$$
\begin{aligned}
V(\pi) - V(\mu_s) &= \gamma^{t_1}\mathbb{E}_{s_{t_1}\sim\rho_{t_1}}\left[Q^{\mu_s}(s_{t_1},\pi(s_{t_1}))\right] - \mathbb{E}_{a_t\sim\mu_s}\left[\sum_{t=t_1}^{T-1}\gamma^t r_t(s_t,a_t)\Big|s_{t_1}\sim\rho_{t_1}\right] \\
&\quad + \sum_{t=t_1+1}^{t_2}\gamma^t\mathbb{E}_{s_t\sim\rho_t}\left[\Delta Q(t)\right] \\
&= \gamma^{t_1}\mathbb{E}_{s_{t_1}\sim\rho_{t_1}}\left[Q^{\mu_s}(s_{t_1},\pi(s_{t_1}))\right] - \gamma^{t_1}\mathbb{E}_{a_t\sim\mu_s}\left[\sum_{t=t_1}^{T-1}\gamma^{t-t_1}r_t(s_t,a_t)\Big|s_{t_1}\sim\rho_{t_1}\right] \\
&\quad + \sum_{t=t_1+1}^{t_2}\gamma^t\mathbb{E}_{s_t\sim\rho_t}\left[\Delta Q(t)\right] \\
&= \gamma^{t_1}\mathbb{E}_{s_{t_1}\sim\rho_{t_1}}\left[Q^{\mu_s}(s_{t_1},\pi(s_{t_1})) - Q^{\mu_s}(s_{t_1},\mu_s(s_{t_1}))\right] + \sum_{t=t_1+1}^{t_2}\gamma^t\mathbb{E}_{s_t\sim\rho_t}\left[\Delta Q(t)\right] \\
&= \sum_{t=t_1}^{t_2}\gamma^t\mathbb{E}_{s_t\sim\rho_t}\left[\Delta Q(t)\right].
\end{aligned}
\tag{28}
$$

From Theorem 2, we know that

$$\left|\Delta Q(t)\right| = \left|Q^{\mu_s}(s_t, \pi(s_t)) - Q^{\mu_s}(s_t, \mu_s(s_t))\right|$$

$$\leq \left[v_M + \gamma\big(k_3 + k_4\big)p_M\right]\big(k_1 + k_2\big)\left|\pi(s_t) - \mu_s(s_t)\right|$$

$$\leq \left[v_M + \gamma\big(k_3 + k_4\big)p_M\right](k_1 + k_2)\xi, \tag{29}$$

where we use $|\pi(s_t) - \mu_s(s_t)| \leq \xi$. Hence, we have

$$\left|V(\pi) - V(\mu_s)\right| = \left|\sum_{t=t_1}^{t_2} \gamma^t \mathbb{E}_{s_t \sim \rho_t}\left[\Delta Q(t)\right]\right|$$

$$\leq \sum_{t=t_1}^{t_2} \gamma^t \mathbb{E}_{s_t \sim \rho_t}\left[\left|\Delta Q(t)\right|\right]$$

$$\leq \sum_{t=t_1}^{t_2} \gamma^t \mathbb{E}_{s_t \sim \rho_t}\left[[v_M + \gamma(k_3 + k_4)p_M](k_1 + k_2)\xi\right]$$

$$= \sum_{t=t_1}^{t_2} \gamma^t \left[v_M + \gamma\big(k_3 + k_4\big)p_M\right]\big(k_1 + k_2\big)\xi$$

$$\leq \xi\gamma^{t_1}\left[v_M + \gamma\big(k_3 + k_4\big)p_M\right]\big(k_1 + k_2\big)\Delta T. \tag{30}$$

So far, we have proved the theorem. In addition, from (30), we can obtain the following two conclusions:

- As $\gamma < 1$, we can see that the later we start explorations (i.e., the larger $t_1$ is), the smaller $|V(\pi) - V(\mu_s)|$ is.

- The larger the exploration time steps $\Delta T$ is, the bigger $|V(\pi) - V(\mu_s)|$ is.

$\square$

# F   Our Approach: SORL Framework

## F.1   Additional Details on SER Policy

### F.1.1   Derivations of the SER Policy $\pi_e$

Recall that the functional optimization problem of the SER policy $\pi_e$ is

$$\max_{\pi_e, \forall s} \quad \mathbb{E}_{a_t \sim \pi_e(\cdot|s_t)}\widehat{Q}(s_t, a_t) \tag{31}$$

$$\text{s.t.} \quad D(\pi_e, \pi_{e,\mathcal{N}}) \leq \epsilon_e, \tag{31a}$$

and the Lagrange function is

$$\mathcal{L}(\pi_e, \lambda) = -\mathbb{E}_{a_t \sim \pi_e(\cdot|s_t)}\widehat{Q}(s_t, a_t) + \lambda(KL(\pi_e, \pi_{e,\mathcal{N}}) - \epsilon_e)$$

$$= \int_{a_t} -\pi_e(a_t|s_t)\widehat{Q}(s_t, a_t)\mathrm{d}a_t + \lambda \int_{a_t} \pi_e \log \frac{\pi_e}{\pi_{e,\mathcal{N}}}\mathrm{d}a_t - \lambda\epsilon_e$$

$$= \int_{a_t} \left[-\pi_e(a_t|s_t)\widehat{Q}(s_t, a_t) + \lambda\pi_e \log \frac{\pi_e}{\pi_{e,\mathcal{N}}}\right]\mathrm{d}a_t - \lambda\epsilon_e$$

$$= \int_{a_t} F[\pi_e(a_t)]\mathrm{d}a_t - \lambda\epsilon_e, \tag{32}$$

where $F[\pi_e(a_t)] = -\pi_e(a_t|s_t)\widehat{Q}(s_t, a_t) + \lambda\pi_e \log \frac{\pi_e}{\pi_{e,\mathcal{N}}}$. According to Euler equation, a necessary condition of the optimal solution to (31) satisfies:

$$\delta\mathcal{L}(\pi_e, \lambda) = \int_{a_t} \frac{\partial F[\pi_e(a_t)]}{\partial \pi_e} \delta\pi_e \mathrm{d}a_t = 0, \quad \lambda \geq 0. \tag{33}$$

Due to the arbitrariness of $\delta\pi_e$, we have $\frac{\partial F[\pi_e(a_t)]}{\partial \pi_e} = 0$, i.e.,

$$-\widehat{Q}(s_t, a_t) + \lambda \log \frac{\pi_e}{\pi_{e,\mathcal{N}}} + \lambda\pi_e \frac{\pi_{e,\mathcal{N}}}{\pi_e} \frac{1}{\pi_{e,\mathcal{N}}} = 0$$

$$\Rightarrow \quad -\widehat{Q}(s_t, a_t) + \lambda \log \frac{\pi_e}{\pi_{e,\mathcal{N}}} + \lambda = 0$$

$$\Rightarrow \quad \exp\{\frac{\widehat{Q}(s_t, a_t)}{\lambda} - 1\} = \frac{\pi_e}{\pi_{e,\mathcal{N}}}$$

$$\Rightarrow \quad \pi_e = \frac{\pi_{e,\mathcal{N}}}{e} \exp\left\{\frac{1}{\lambda}\widehat{Q}(s_t, a_t)\right\}. \tag{34}$$

To ensure that $\pi_e$ is a distribution over actions, we modify it to

$$\pi_e = \frac{1}{C(s_t)} \pi_{e,\mathcal{N}} \exp\left\{\frac{1}{\lambda}\widehat{Q}(s_t, a_t)\right\}, \tag{35}$$

where $C(s_t) = \int_{a_t} \frac{1}{\sqrt{2\pi\sigma^2}} \exp\{-\frac{(a_t - \mu_s(s_t))^2}{2\sigma^2} + \frac{1}{\lambda}\widehat{Q}(s_t, a_t)\}\mathrm{d}a_t$ acts as the normalization factor. Note that the KL divergence $KL(\pi_e, \pi_{e,\mathcal{N}})$ we used in the derivations is formulated as $\int_{a_t} \pi_e \log \frac{\pi_e}{\pi_{e,\mathcal{N}}}\mathrm{d}a_t$ rather than $\int_{a_t} \pi_{e,\mathcal{N}} \log \frac{\pi_{e,\mathcal{N}}}{\pi_e}\mathrm{d}a_t$. The reason is that: the exploration policy will be calculated as $\pi_e = \pi_{e,\mathcal{N}} \frac{\widehat{Q}(s_t, a_t)}{\lambda}$ if we use the latter KL divergence, which cannot guarantee the non-negative property of $\pi_e$. We also note that the form of SER policy $\pi_e$ here resembles the results derived in [28, 29, 14]. Nonetheless, we view the problem as a functional optimization problem and utilize the Euler equation to obtain the results.

### F.1.2 Practical Implementations of $\pi_e$

As the Q function $\widehat{Q}$ is a neural network, we cannot directly sample actions from $\pi_e$. Nonetheless, we can obtain the value of $\pi_e$ of each action $a_t$ given a state $s_t$. Hence, we uniformly sample $M \in \mathbb{N}_+$ actions $\{a_t^m\}_{m=1}^M$ within the safety zone $[\mu_s(s_t) - \xi, \mu_s(s_t) + \xi]$, and the possibility of selecting action $a_t^m$ is calculated as $\pi_e(a_t^m|s_t)/\sum_{m=1}^M \pi_e(a_t^m|s_t)$. Then we sample the actions from $\{a_t^m\}_{m=1}^M$ for explorations.

### F.1.3 More on the Safety Requirement

In other realms, such as robotics, it is possible to construct an *immediate-evaluated* safety function to evaluate the safety of current state-action pairs, which is merely related to the values of them. For example, a robot can instantly be in danger due to an action at a state, for example, dashing against the wall. However, it is not appropriate to construct such kind of safety functions in auto-bidding. Generally, there are usually two main kinds of dangerous situations in auto-bidding:

- the first situation is the extremely quick burns of budgets with high cost-per-action (CPA) values, which is probably caused by continuously bidding at very high prices;
- the second situation is the extremely slow consumptions of budgets, which is probably caused by continuously bidding at very low prices.

Both of these two dangerous situations cannot be attributed to a specific state-action pair, but to a long-term auto-bidding policy. In fact, any action (bids) in any state would be safe as long as the total subsequent rewards maintains at a high level. For example, bidding an oddly high price in time step $t$, but bidding at reasonable prices afterwards and the overall reward at the end of the episode is at a high level would be acceptable. On the contrary, bidding reasonably at present moment but continuously bidding at oddly high prices afterwards, resulting in a low accumulated reward at the end of the episode, would harm the interests of advertisers and not be safe in auto-bidding.

## F.2  Additional Details on V-CQL

### F.2.1  Nearly Quadratic Form of Q Functions

Fig. 6 shows the optimal Q functions in the simulated experiments, and Fig. 7 shows the Q function of the state-of-the-art Q functions. We can see that the Q functions are all in quadratic forms. Hence, based on the proved Lipschitz smooth property of Q functions, we can reasonably assume that the optimal Q function is nearly quadratic.

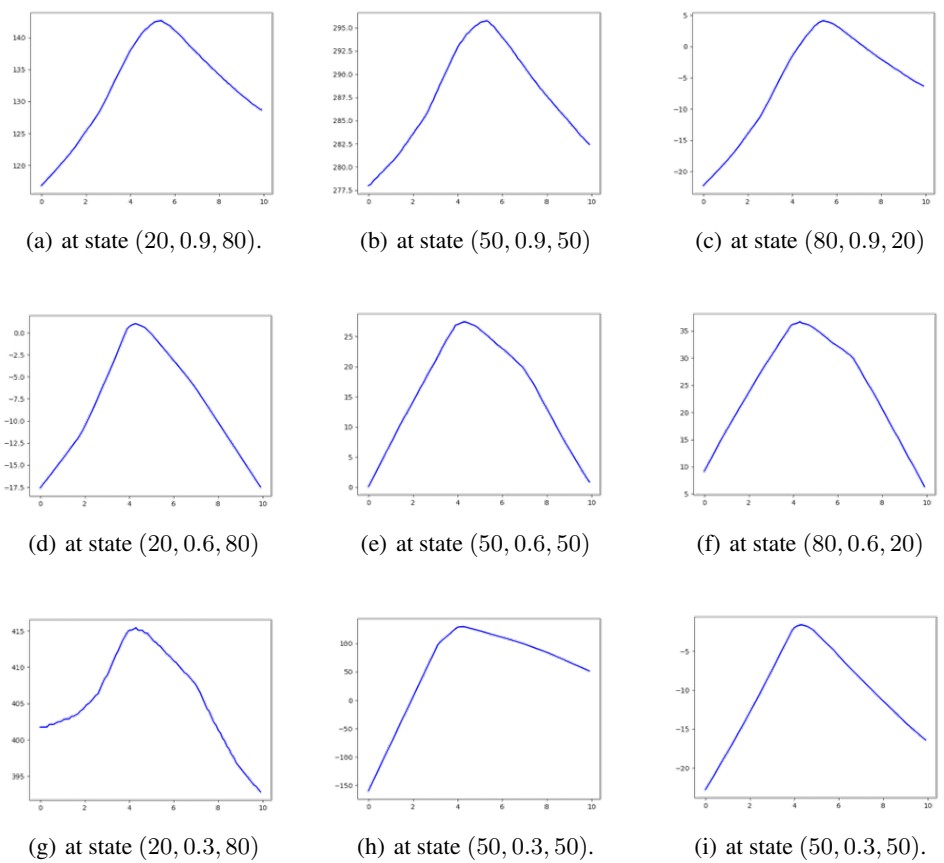

Figure 6: The form of optimal Q functions in the simulated experiments are all nearly in quadratic form. In this example, the total budget is 100, and we choose time left to be $0.9, 0.6, 0.3$, respectively.

### F.2.2  Complete V-CQL Method

In this subsection, we specify the novelties of the proposed V-CQL algorithm and analyze its advantages and relations to previous offline RL methods.

**CQL and its variants.** Recall that general CQL [13] algorithm (i.e., CQL($\mathcal{R}$)) can be expressed as

$$\min_Q \; -\alpha \underbrace{\mathbb{E}_{s_k \sim \mathcal{D}, a_k \sim \widehat{\pi}_\beta}[\widehat{Q}(s_k, a_k)]}_{\text{Reward the in-distribution actions}} + \frac{1}{2} \underbrace{\mathbb{E}_{s_k, a_k, s'_k \sim \mathcal{D}}\left[\left(\widehat{Q}(s_k, a_k) - \widehat{\bar{B}}\bar{Q}(s_k, a_k)\right)^2\right]}_{\text{\textbf{Bellman error:} minimizing TD error}} +$$

$$\underbrace{\max_\mu \; \alpha \mathbb{E}_{s_k \sim \mathcal{D}, a \sim \mu}[\widehat{Q}(s_k, a)] + \underbrace{\mathcal{R}(\mu)}_{\text{regularizer}}}_{\text{choose } \mu \text{ to maximize the current Q-function}}, \tag{36}$$

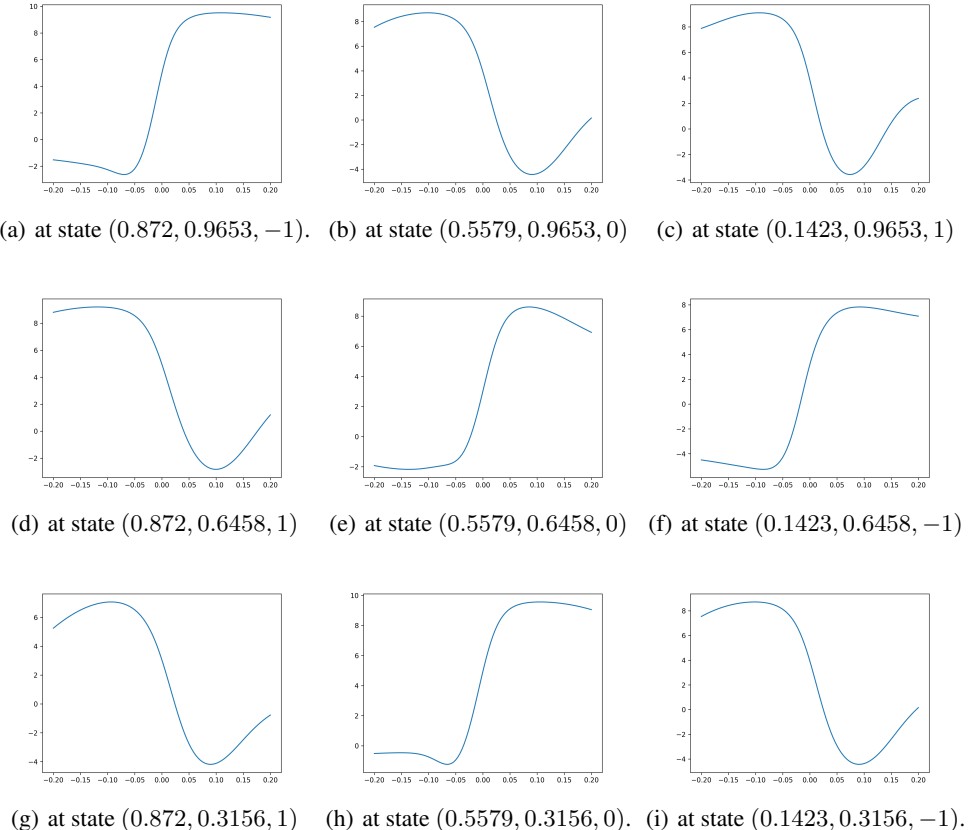

(a) at state $(0.872, 0.9653, -1)$.  (b) at state $(0.5579, 0.9653, 0)$  (c) at state $(0.1423, 0.9653, 1)$

(d) at state $(0.872, 0.6458, 1)$  (e) at state $(0.5579, 0.6458, 0)$  (f) at state $(0.1423, 0.6458, -1)$

(g) at state $(0.872, 0.3156, 1)$  (h) at state $(0.5579, 0.3156, 0)$.  (i) at state $(0.1423, 0.3156, -1)$.

Figure 7: The form of Q functions of the auto-bidding policy trained by USCB based on real-world dataset are all nearly in quadratic form. Note that the state have already been normalized, and we choose time left to be $0.9653, 0.6458, 0.3156$, respectively.

where $\mathcal{D}$ denotes the offline dataset, $\widehat{\pi}_\beta \triangleq \frac{\sum_{s_k, a_k \sim \mathcal{D}} \mathbf{1}[s=s_k, a=a_k]}{\sum_{s_k \sim \mathcal{D}} \mathbf{1}[s=s_k]}$ is the estimated behavior policy based on $\mathcal{D}$, and $\bar{\mathcal{B}}$ represents the Bellman operator. There are two popular variants of CQL, including CQL($\mathcal{H}$) and CQL($\rho$). They both implement the regularizer as a KL-divergence between $\mu$ and a prior distribution $\rho$. Specifically,

- CQL($\mathcal{H}$) chooses $\rho$ to be a uniform policy, i.e., $\mathcal{R}(\mu) = -D_{\mathrm{KL}}(\mu, \mathrm{Unif}(a))$. Hence, it turns the third term in (36) into a *conservative penalty*.

- CQL($\rho$) chooses $\rho$ to be the previous policy $\widehat{\pi}^{k-1}$, i.e., $\mathcal{R}(\mu) = -D_{\mathrm{KL}}(\mu, \widehat{\pi}^{k-1})$. Hence, it turns the third term in (36) into both a *policy constraint* and a *conservative penalty*.

**V-CQL.** The novelties of the proposed V-CQL algorithm are in three-fold. Firstly, as we know the exact formulations of behavior policies generating the data in the offline dataset $\mathcal{D}$ (i.e., the data in $\mathcal{D}_s$ is generated by $\mu_s$, and the data in $\mathcal{D}_{on,\tau}$ is generated by $\pi_{e,\tau}$), we can substitute the $\widehat{\pi}_\beta$ in (36) directly by the behavior policies. This cuts down the estimations process of behavior policy $\widehat{\pi}_\beta$. For convenience, we uniformly denote the behavior policies as $\mu_b$, where

$$\mu_b = \begin{cases} \mu_s, & \text{for data in } \mathcal{D}_s, \\ \pi_{e,\tau}, & \text{for data in } \mathcal{D}_{on,\tau}. \end{cases} \tag{37}$$

Secondly, we adapt the policy constraint in CQL($\rho$) to a constraint on the Q function. Specifically, we devise the regularizer $\mathcal{R}(\mu)$ as (8) in the manuscript. Note that, as $\mathcal{R}(\mu)$ is not a function of $\mu$, the maximizing operation in the third term of (36) is not needed. This way of policy constraint can reduce the performance variance compared to CQL($\mathcal{H}$), and has more flexibilities than the policy constraint in CQL($\rho$) as well as other form of policy constraints direct on policies (such as BCQ). Thirdly, the

policy $\rho$ in CQL($\rho$) utilizes the policy in the previous training iterations. The $\widehat{Q}_{\text{old}}$ in $\mathcal{R}(\mu)$ of the V-CQL also leverages a previous Q function. Nonetheless, $\widehat{Q}_{\text{old}}$ does not change during the whole training process at iteration $\tau$ and keeps $\widehat{Q}_{\tau-1}$ until the next iteration. Fig. 8 shows the difference between the V-CQL and CQL($\rho$). Besides, we adopt the conservative penalty in the CQL($\mathcal{H}$) in the V-CQL method. Overall, the V-CQL algorithm can be expressed as

$$
\min_{Q} \quad \alpha_1 \underbrace{\mathbb{E}_{s_k \sim \mathcal{D}}\left[\log \sum_{a \sim \text{Unif}(\mathcal{A})} \exp(\widehat{Q}(s_k, a))\right]}_{\textbf{conservative penalty: } \text{punishing all actions}} - \alpha_2 \underbrace{\mathbb{E}_{s_k \sim \mathcal{D}}\left[\widehat{Q}(s_k, \mu_b(s_k))\right]}_{\text{Reward the in-distribution actions}}
$$

$$
+ \frac{1}{2} \underbrace{\mathbb{E}_{s_k, a_k, s_k' \sim \mathcal{D}}\left[\left(\widehat{Q}(s_k, a_k) - \bar{\mathcal{B}}\bar{Q}(s_k, a_k)\right)^2\right]}_{\textbf{Bellman error: } \text{minimizing TD error}}
$$

$$
+ \beta \, \mathbb{E}_{s_k \sim \mathcal{D}}\underbrace{\left[D_{\text{KL}}\left(\frac{\exp(\widehat{Q}(s_k, \cdot))}{\sum_{a \sim \text{Unif}(\mathcal{A})} \exp(\widehat{Q}(s_k, a))}, \frac{\exp(\widehat{Q}_{\text{qua}}(s_k, \cdot))}{\sum_{a \sim \text{Unif}(\mathcal{A})} \exp(\widehat{Q}_{\text{qua}}(s_k, a))}\right)\right]}_{\textbf{policy constraint: } \text{constraining the distribution shifts of the Q function}}, \quad (38)
$$

where $\alpha_1, \alpha_2, \beta > 0$ are constants, $\bar{\mathcal{B}}$ denotes the Bellman operator, and $\bar{Q}$ is the target Q function. Note that we also randomly sample the actions from the whole action space to calculate the KL-divergence between the old and new Q functions. At iteration $\tau$, the Q function $\widehat{Q} \leftarrow \widehat{Q}_\tau$ is trained based on $\widehat{Q}_{\text{qua}} \leftarrow \widehat{Q}_{\tau-1}$. In practice, the V-CQL can be applied to either Q learning RL algorithms with implicit policies, such as DQN, or actor-critic RL algorithms with explicit policies, such as DDPG. In the SORL, we leverage the DDPG method to train explicit auto-bidding policies, where the Q functions are trained by the V-CQL.

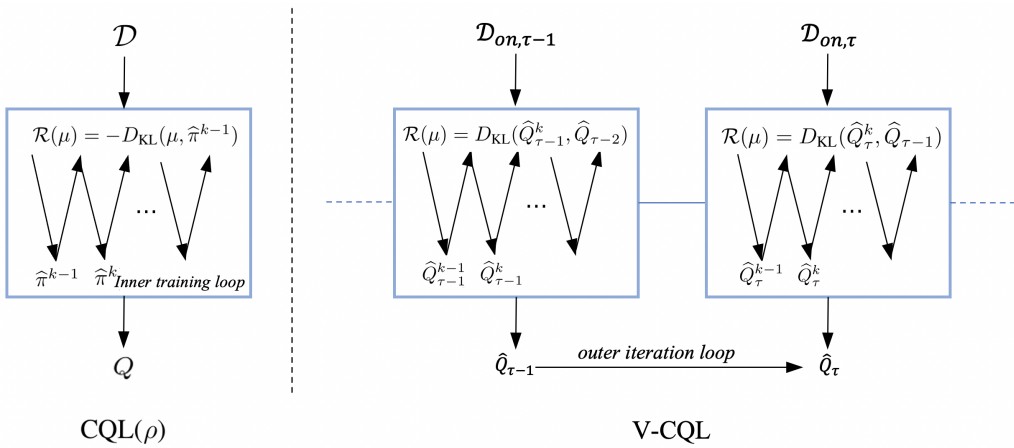

Figure 8: The differences between the V-CQL and CQL($\rho$).

**V-CQL combines the advantages of conservative penalty and policy constraint.** As stated in the Appendix B, there are two main ways to mitigate the extrapolation errors in offline RL methods, including the conservative penalty where explicit punishments are imposed to Q functions (a typical method is the CQL($\mathcal{H}$) [13]), and the policy constraints where the KL-divergence between the trained policy and the original policy is limited within a certain range (BCQ [11] imposes constraint directly on the policy, while CQL($\rho$) imposes the constraint on the Q function). The first way has the potential to train policies with high performance, but can have high performance variance. The second way can have low performance variance since it directly imposes the constraints on the policies. However, it generally cannot achieve the performance as good as the first way [26]. Besides, imposing constraints directly on policies (as BCQ does) in the SORL framework will face many challenges. For example, the behavior policy of the offline data $\mathcal{D}$ is mixed policy since $\mathcal{D}$ is composed of data collected by different policies. One needs to train a new perturbation model $\xi_\phi$ at each iteration $\tau$ for each exploration policy $\pi_{e,\tau}$ and cannot utilize the dataset in previous iterations. As the exploration policy

$\pi_{e,\tau}$ does not equal to the auto-bidding policy $\mu_{\tau-1}$ in the previous iteration $\tau - 1$, the auto-bidding policy cannot be iteratively improved. Nonetheless, the proposed V-CQL combines the advantages of both conservative penalty and policy constraint methods: the V-CQL can reduce the auto-bidding policy's performance variance and iteratively improves the auto-bidding policy in an elegant way.

## F.3 SORL Framework Pseudocode

The overall SORL framework algorithm is presented in Algorithm 1. Specifically, the SORL works in an iterative manner. In each iteration $\tau$, the SORL collects data directly from the RAS with the proposed exploration policy $\pi_{e,\tau}$ and use the V-CQL method to train the auto-bidding policy $\mu_\tau$ with the newly collected data. Note that in the first (i.e., 0-th) iteration, we need a known policy to start the data collection process, and thus, boot the SORL. As the policy will directly interact with the RAS, it should be safe. Hence, we leverage the state-of-the-art auto-bidding policies, for example, USCB [5] that has already been deployed to the RAS in practice, to make a warm booting. As proved in the manuscript, the subsequent exploration policies $\pi_{e,\tau}$ is guaranteed to be safe. Due to the constantly feedback from the collected data, the auto-bidding policy $\mu_\tau$ will be improved.

---

**Algorithm 1** SORL Framework

---

**Inputs:** The initial safety policy $\mu_s$.
**Outputs:** The auto-bidding policy $\mu^*$ and its Q function $Q^*$.
**Warm Booting:** Collect data $\mathcal{D}_s$ from the RAS with the safe policy $\mu_s$, and train the auto-bidding policy $\mu_0$ and $\widehat{Q}_0$. Let $\tau \leftarrow 1$.
**Iteration Process:**

1: **while** $\widehat{Q}_\tau$ not convergence **do**
2:     Construct the SER policy $\pi_{e,\tau}$ based on $\mu_s$ and $\widehat{Q}_\tau(s_t, a_t)$;
3:     Explore in the RAS with $\pi_{e,\tau}$ and collect the dataset $\mathcal{D}_{on,\tau}$.
4:     Train the new auto-bidding policy $\mu_\tau$ and its Q function $\widehat{Q}_\tau$ with V-CQL based on the collected data.
5:     $\tau \leftarrow \tau + 1$.
6: **end while**
7: Let $Q^* \leftarrow \widehat{Q}_\tau$, and $\mu^* \leftarrow \mu_\tau$.

---

# G    Experimental Results

In this section, we present additional information on the experiment parameters and OPE method in auto-bidding. In addition, we conduct extra experiments to validate the effectiveness of our approach.

## G.1    Experiment Setup

**Simulated Advertising System.** We conduct experiments on the s-RAS mentioned in Appendix A.2. The parameters of the simulated advertising system are shown in Table. 4. We implement the V-CQL method by the actor-critic framework. The hyper-parameters used in the RL training are summarized in Table. 3.

**Real-world Advertising System (RAS).** We conduct experiments on one the world's largest E-commerce platforms, TaoBao. We apply the SORL framework to thousands of real advertisers from April 28, 2022 to May 26, 2022 to validate the effectiveness of it. We implement the V-CQL method by the actor-critic framework, whose hyper-parameters are summarized in Table. 5.

Table 3: The hyper-parameters of DDPG in experiments on the s-RAS.

| Hyper-parameters | Values |
|---|---|
| Optimizer | Adam |
| Learning rate for critic network | $1 \times 10^{-4}$ |
| Learning rate for actor network | $1 \times 10^{-4}$ |
| Soft updated rate | 0.01 |
| Buffer size | 1000 |
| Sampling size | 200 |
| Discounted factor $\gamma$ | 0.99 |
| $\alpha_1, \alpha_2$ for V-CQL | 0.002 |
| $\beta$ for V-CQL | 0.001 |
| $\sigma$ in SER policy $\pi_\tau$ | 1 |
| $\lambda$ in SER policy $\pi_\tau$ | 0.1 |
| sample numbers of SER policy $\pi_\tau$ | 1000 |
| $\xi$ sample range of SER policy $\pi_\tau$ | 0.5 |

Table 4: The parameters used in the s-RAS.

| Parameters | Values |
|---|---|
| Number of advertisers | 100 |
| Time steps in an episode, $T$ | 96 |
| Minimum number of impression opportunities $N_t$ | 100 |
| Maximum number of impression opportunities $N_t$ | 500 |
| Minimum budget | 100,000 Yuan |
| Maximum budget | 200,000 Yuan |
| Value of impression opportunities in stage 1, $v_{j,t}^1$ | $0 \sim 1$ |
| Value of impression opportunities in stage 2, $v_{j,t}^2$ | $0 \sim 1$ |
| Minimum bidding price, $A_{\min}$ | 0 Yuan |
| Maximum bidding price, $A_{\max}$ | 1,000 Yuan |
| Maximum value of impression opportunity, $v_M$ | 1 |
| Maximum market price, $p_M$ | 1,000 Yuan |

## G.2 Additional Results

### G.2.1 Ablation Study: Compare the V-CQL with BCQ, CQL($\mathcal{H}$) and CQL($\rho$)

In the manuscript, we compare the V-CQL with the CQL method, specifically CQL($\mathcal{H}$). Here, we compare the V-CQL with more offline RL methods in the RAS, which can serve as an ablation study. As shown in Table. 6, the V-CQL outperforms the BCQ, CQL($\mathcal{H}$) and CQL($\rho$) in all metrics in the RAS. Firstly, the comparison with BCQ and CQL($\mathcal{H}$) indicates that the V-CQL combines the advantages of conservative penalty and policy constraint. Secondly, the comparison with the CQL($\rho$) validates the effectiveness of the proposed form of policy constraint (8) in the manuscript.

### G.2.2 Affects of Hyper-parameters $\sigma$ and $\lambda$ on SER Policy $\pi_\tau$

We apply the SER policy $\pi_e$ to the s-RAS with different hyper-parameters $\sigma$ and $\lambda$, and the total accumulated rewards (Q value) are shown in Fig. 9. Specifically, Fig. 9(a) shows the accumulated rewards of $\pi_e$ constructed by the Q function $\widehat{Q}_{(1)}(s_t, a_t)$ that has poor performance (with ROI of 3.39), while Fig. 9(b) shows the accumulated rewards of $\pi_e$ constructed by the Q function $\widehat{Q}_{(7)}(s_t, a_t)$ that has good performance (with ROI of 3.82). The initial safe policy $\mu_s$ has a total accumulated reward of 212.36 and a ROI of 3.64. We can see that the declines in Q values of $\pi_e$ under all hyper-parameters are within 5% with respect to the Q value of $\mu_s$. This indicates the safety of the SER policy. Moreover, from Fig. 9(a), we can see that when the Q function of $\pi_e$ has poor performance,

Table 5: The hyper-parameters of DDPG when applying the SORL to the RAS.

| Hyper-parameters | Values |
|---|---|
| Optimizer | Adam |
| Learning rate for critic network | $2 \times 10^{-5}$ |
| Learning rate for actor network | $2 \times 10^{-5}$ |
| Soft updated rate | 0.01 |
| Buffer size | $\sim 10000$ |
| Sampling size | 64 |
| Discounted factor $\gamma$ | 0.999 |
| $\alpha_1, \alpha_2$ for V-CQL | 0.001 |
| $\beta$ for V-CQL | 0.002 |
| $\sigma$ in SER policy $\pi_\tau$ | 0.15 |
| $\lambda$ in SER policy $\pi_\tau$ | 0.3 |
| sample numbers of SER policy $\pi_\tau$ | 50 |
| $\xi$ sample range of SER policy $\pi_\tau$ | 0.1 |

Table 6: Ablation study in the RAS: compare V-CQL with BCQ (policy constraint), CQL($\mathcal{H}$) (conservative penalty) and CQL($\rho$) (both policy constraint and conservative penalty).

| Methods | no conservative penalty: V-CQL vs. BCQ | | | |
|---|---|---|---|---|
| | BuyCnt | ROI | CPA | ConBdg |
| BCQ | 8,746 | 3.78 | 23.85 | 208,576.53 |
| V-CQL | 8,992 | 3.94 | 23.48 | 211,142.59 |
| variation | **+2.81%** | **+4.23%** | **-1.54%** | **+1.23%** |
| Methods | no policy constraint: V-CQL vs. CQL($\mathcal{H}$) | | | |
| | BuyCnt | ROI | CPA | ConBdg |
| CQL($\mathcal{H}$) | 40,462 | 3.87 | 21.42 | 845,621.15 |
| V-CQL | 42,236 | 3.95 | 20.29 | 856,913.14 |
| variation | **+4.38%** | **+2.07%** | **-5.27%** | **+1.33%** |
| Methods | with different versions of policy constraint : V-CQL vs. CQL($\rho$) | | | |
| | BuyCnt | ROI | CPA | ConBdg |
| CQL($\rho$) | 9,523 | 4.04 | 20.99 | 199,873.22 |
| V-CQL | 9,867 | 4.20 | 20.30 | 200,291.00 |
| variation | **+3.61%** | **+3.96%** | **-3.28%** | **+0.21%** |

larger $\lambda$ and smaller $\sigma$ can make the SER policy $\pi_e$ safer. This is because $\widehat{Q}_{(1)}(s_t, a_t)$ does not lead the explorations to a good direction, and stick to the safe policy $\mu_s$ can be a safer choice. On the contrary, from Fig. 9(b), we can see that when the Q function has good performance, smaller $\lambda$ can make $\pi_e$ more safer, and $\sigma$ can be set to a larger value to increase the exploration efficiency.

### G.2.3 Complete Results of Table. 2 in the Manuscript

The complete experiment results of Table. 2 in the manuscript are shown in Table 8.

### G.2.4 Compare With Multi-agent Auto-bidding Algorithms

Some researchers may consider the comparison between our approach and the multi-agent auto-bidding algorithms. However, we claim that the problem setting of this paper is different from that of the multi-agent algorithms [36]. As we stated in Section 3, the problem we considered is how to maximize the total value of a single advertiser, which is naturally a single-agent problem.

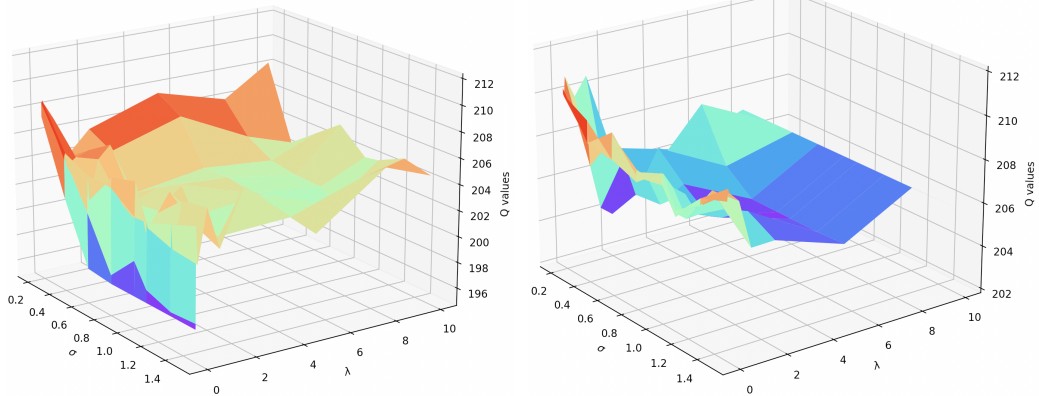

(a) Q value of $\pi_e$ with Q function $\widehat{Q}_{(1)}(s, a)$ that has poor performance.

(b) Q value of $\pi_e$ with Q function $\widehat{Q}_{(7)}(s, a)$ that has good performance.

Figure 9: The accumulated rewards of the SER policy $\pi_e$ constructed by Q functions with different performance level under different hyper-parameters $\sigma$ and $\lambda$.

What we argue in Fig. 2 is that all other advertisers acting as a part of the environment are not correctly modeled in the VAS, which will cause the IBOO. In addition, our method can solve the IBOO, including the inaccurate market price changing issue. However, the multi-agent auto-bidding problem [36] studies how to realize multi-objective goals, involving the interests of advertisers and the platform. Hence, it is not very proper to compare our algorithm with the multi-agent algorithms [36] that solves a different problem.

Nonetheless, we conduct real-world A/B test between our approach and the multi-agent algorithm in [36], and the results are shown in Table. 7. We can see that the SORL largely outperforms than the multi-agent algorithm in the performance indexes considered in the single-agent auto-bidding problem in our paper.

Table 7: Real-world A/B tests between the SORL and the multi-agent auto-bidding algorithm.

| Algorithms | BuyCnt | ROI | CPA | ConBdg |
|---|---|---|---|---|
| multi-agent method [36] | $121,616$ | 2.79 | 44.86 | $5,455,507.35$ |
| SORL | **139,599** | **3.15** | **40.05** | **5,590,858.50** |
| variations | **+14.79%** | **+12.90%** | **-10.72%** | **+2.48%** |

# H   Broader Impact

In this paper, we propose a SORL framework to improve the state-of-the-art auto-bidding policies with direct explorations in the real-world advertising system (RAS). To the best of our knowledge, we are the first to systematically analyze the IBOO problem in auto-bidding and complete resolve it with an online RL manner. The derived auto-bidding policy can benefit both the advertisers and the companies at the same time, which can generate huge social and economic benefits. We believe that the proposed SORL framework will be the next generation of auto-bidding paradigm. In addition, the IBOO problem does not only exists in the auto-bidding. In fact, it resembles the sim2real problem in many other realms, such as robotics. The proposed SORL framework is a general method which can be easily applied to other applications to solve the sim2real problem.

Table 8: The complete experiment results of SORL framework in the RAS.

| Metrics | A/B Tests | Auto-bidding policy $\mu_\tau$ derived in iteration $\tau$ | | | |
| --- | --- | --- | --- | --- | --- |
| | | 0-th: $\mu_0$ | 1-th: $\mu_1$ | 2-th: $\mu_2$ | 3-th: $\mu_3$ |
| BuyCnt | auto-bidding policy $\mu_{\tau-1}$ | 40,926.00 | 7,982.00 | 8,571.00 | 9,207.00 |
| | | 42,236.00 | 8,034.00 | 8,611.00 | 9,295.00 |
| | | **+3.20%** | **+0.65%** | **+0.47%** | **+0.95%** |
| | USCB (state-of-the-art) | 40,926.00 | 6,358.00 | 7,432.00 | 9,358.00 |
| | | 42,236.00 | 6,575.00 | 7,697.00 | 9,709.00 |
| | | **+3.20%** | **+3.41%** | **+3.57%** | **+3.75%** |
| ROI | auto-bidding policy $\mu_{\tau-1}$ | 3.90 | 3.58 | 3.88 | 3.20 |
| | | 3.95 | 3.65 | 3.89 | 3.31 |
| | | **+1.28%** | **+1.96%** | **+0.26%** | **+3.20%** |
| | USCB (state-of-the-art) | 3.90 | 3.47 | 3.76 | 3.47 |
| | | 3.95 | 3.57 | 3.82 | 3.55 |
| | | **+1.28%** | **+2.88%** | **+1.60%** | **+2.48%** |
| CPA | auto-bidding policy $\mu_{\tau-1}$ | 20.71 | 21.32 | 22.38 | 23.21 |
| | | 20.29 | 21.05 | 22.35 | 23.44 |
| | | **-2.01%** | **-1.27%** | **-0.13%** | **-1.01%** |
| | USCB (state-of-the-art) | 20.71 | 22.52 | 20.31 | 24.52 |
| | | 20.29 | 22.31 | 20.11 | 23.60 |
| | | **-2.01%** | **-0.93%** | **-0.98%** | **-3.91%** |
| ConBdg | auto-bidding policy $\mu_{\tau-1}$ | 847,403.12 | 170,176.24 | 191,818.98 | 215,695.14 |
| | | 856,913.14 | 169,115.70 | 192,455.85 | 215,828.40 |
| | | **+1.12%** | -0.62% | **+0.33%** | **+0.06%** |
| | USCB (state-of-the-art) | 847,403.12 | 143,182.16 | 150,943.92 | 229,492.24 |
| | | 856,913.14 | 146,688.25 | 154,786.67 | 229,141.06 |
| | | **+1.12%** | **+2.45%** | **+2.55%** | -0.15% |