# OpenReview forum: "Sustainable Online Reinforcement Learning for Auto-bidding"
_NeurIPS.cc/2022/Conference — NeurIPS 2022 Accept_

### Official Review · Reviewer_o6ML · 2022-07-10

**Rating:** 7
**Confidence:** 4
**Soundness:** 3 good
**Presentation:** 3 good
**Contribution:** 3 good

**Summary:**

This paper tries to address the problem in auto-bidding that training can only be carried out offline due to safety concerns. The performance of the learned policy could deteriorate due to the difference between the offline simulation and online real environment. The authors propose a sustainable online reinforcement learning paradigm which uses a safe policy to explore the real enviroment.

**Questions:**

(1) What do you mean by $p_{j,t}$ and $v_{j,t}$ in proposition 2? Do you mean $p_{j,t}^2$ and $v_{j,t}^2$?
(2) What do you mean by $\mu_s$ in Figure 6?
(3) Does the $\pi_e$ policy mean the SER policy after 7 iterations?
(4) Why do you only show results after three iterations in Table 3?

**Limitations:**

No such issues are identified in the paper.

**Strengths And Weaknesses:**

Strengths:
(1) the problem to be addressed is important and novel.
(2) the proposed idea is interesting and technical depth is deep.
(3) experiments are conducted on a real world platform.

Weaknesses (The revised paper and the authors' response addressed these major concerns.):
(1) There are approximations/relaxations without proper justification or theoretical guarantee.
(2) There could be flaws in some proofs.
(3) The experiments fail to compare the proposed method to SOTA methods.

Detailed comments:

The major problems with this work are:

-There are approximations/relaxations without proper justification or theoretical guarantee. At the end of section 3, the authors said we can relax the constraint "the safty zone is effective only when the subsequent actions strictly follow the saft policy $\mu_s$". But there is neither jutification for the relaxation nor theoretical analysis on how this relaxation could affect the safe property of the online exploration. Another issue lies in section 4.3, where the authors modify the SER policy without justification.

-The proof for theorem 1 seems problematic. The upper bound in Eq.(12) could be violated by such a case: $\frac{p_{j,t}^2}{v_{j,t}^2} > a_2, \forall j \in \{1, \dots, N_t\}$, which means $a_2$ fails to win any impressions. In this case, subtracting the winning impression upperbound for $a_2$ seems to underestimate the bound.

-The authors fail to compare to the SOTA works in the auto-bidding field, e.g., ["Multi-Agent Cooperative Bidding Games for Multi-Objective Optimization in e-Commercial Sponsored Search." SIGKDD 2021] develops an evolutionary strategy to learn the optimal policy and achieves better performance than RL methods. Furthermore, as the authors argue in Figure 2(b), the bidding involving all advertisers naturally falls into the realm of multi-agent RL. However, the proposed method is still a single agent solution which cannot well handle the market price change issue. In comparison, the above work sets up a multi-agent bidding game for policy learning.


Minor issues:

-The presentation could be improved. The figures are not detailedly explained in the main text. Some notations are confusing. See the first question in the question part for an example. Another example is that the authors use both $\pi$ and $\mu$ to denote a policy.

-Writing could be improved. There are noticeable language errors, e.g., "derivation"-->"deviation", "are in four-fold", "is substitute by", etc.

---

> ### Author Response · Authors · 2022-08-02
> **Response to Reviewer o6ML**
>
> Thank you for the in-depth evaluation of our work and for the constructive feedback. We address the concerns that the reviewer stated point by point as follows.
>
> **1. There are approximations/relaxations without proper justification or theoretical guarantee.**
>
> Thank the reviewer for this concern. Firstly, we claim that the method without relaxations still works for the problem we considered, and the theoretical guarantees are given in the original manuscript.
> Then, to make the theory more general, we have added the theoretical guarantees and corresponding proofs to all the relaxations the reviewer mentioned in the revised manuscript. The main contents are summarized as follows.
>
> **(1.1) Issue at the end of Section 3**: We have added the theoretical guarantee to this relaxation in Theorem 3 in the revised manuscript, and presented the detailed proofs in Appendix E.3 in the revised manuscript.
> Specifically, consider a relaxed exploration policy $\pi$, i.e., taking actions by sampling within the safety zone $[\mu_s-\xi,\mu_s+\xi]$ in $\Delta T\ge1$ consecutive time steps (starting from time step $t_1$ and ending at time step $t_2$, where $\Delta T=t_2-t_1+1$) and following the safe policy $\mu_s$ in the rest $T-\Delta T$ time steps.
> Denote the expected accumulative value of impression opportunity the advertiser earns when using $\pi$ and $\mu_s$ as $V(\pi)$ and $V(\mu_s)$, respectively.
> We prove that the difference between $V(\pi)$ and $V(\mu_s)$ is not bigger than a certain upper bound:
> \begin{align}
> 	\bigg\vert V(\pi) - V(\mu_s)\bigg\vert\le \xi\gamma^{t_1}\bigg[v_M+\gamma\big(k_3+k_4\big)p_M\bigg]\big(k_1+k_2\big)\Delta T=\xi\gamma^{t_1}L_Q\Delta T\le\epsilon_s\gamma^{t_1}\Delta T,
> \end{align}
> where we use $\xi\le\frac{\epsilon_s}{L_Q}$.
> We can also derive the upper bound of the difference between the safety functions
> \begin{align}
> 	{
> 	\bigg\vert Q^{\pi}(s_{t_1},\pi(s_{t_1}))-Q^{\mu_s}(s_{t_1},\mu_s(s_{t_1}))	\bigg\vert =\frac{1}{\gamma^{t_1-1}}	\bigg\vert V(\pi)-V(\mu_s)	\bigg\vert\le\epsilon_s\gamma\Delta T.}
> \end{align}
> Moreover, if we wish the decrease of $V(\pi)$ relative to $V(\mu_s)$ by no more than $\epsilon_s'>0$, then we can redesign $\xi$ as $\xi\le \frac{\epsilon_s'}{\gamma^{t_1}L_Q\Delta T}$.
> In addition, we have conducted both simulated and real-world experiments in Section 5.1 and Appendix G.3.3 to validate the safety of the exploration policy in the original manuscript.
>
> **(1.2) Issue lies in Section 4.3**: The exploration policies, including $\pi_{e,\mathcal{N}}$, SER $\pi_e$, and modified SER $\pi_{e,\tau}$, are all specific implementations of the relaxed exploration policy $\pi$. The differences between them are only the way to sample within the safety zone $[\mu_s-\xi,\mu_s+\xi]$. Hence, they all satisfy the safety property. Specifically, when $\xi\le\frac{\epsilon_s}{L_Q}$, the modified SER satisfies $|V(\pi_{e,\tau})-V(\mu_s)|\le\epsilon_s\gamma^{t_1}\Delta T, \forall \tau$ and $|Q^{\pi_{e,\tau}}(s_{t_1},\pi(s_{t_1}))-Q^{\mu_s}(s_{t_1},\mu_s(s_{t_1}))|\le \epsilon_s\gamma\Delta T$.
>
> **2. The proof for theorem 1 seems problematic.**
>
> Thank you for this concern. We have carefully examined the derivations of Theorem 1, especially Eq.(12), and we believe there is no flaw in the proofs.
> Firstly, Theorem 1 tries to calculate the upper bound for $\frac{|r_t(s_t,a_1)-r_t(s_t,a_2)|}{|a_1-a_2|}$ and there is no subtraction for "winning impression upper bound for $a_2$" in the proof. Secondly, the upper bound in Eq.(12) will not be violated by the case $\frac{p^2_{j,t}}{v^2_{j,t}}>a_2,\forall j\in\{1,2,...,N_t\}$. This is because "$a_2$ falls to win any impressions" has already been included in the considered cases presented in the original manuscript:
>
> <1> impressions lost with $a_2$ in stage 1, and
>
> <2> impressions won with $a_2$ in stage 1 but lost in stage 2.
>
> The upper bounds for |Number of impressions won with $a_1$ - Number of case <1>|, and |Number of impressions won with $a_1$ - Number of case <2>| are calculated in Eq. (10) and Eq.(11), respectively, and are summed up in Eq.(12). Hence, the upper bound in Eq.(12) is robust.

---

> > ### Author Response · Authors · 2022-08-02
> > **Response to Reviewer o6ML**
> >
> > **3. The authors fail to compare to the SOTA works in the auto-bidding field.**
> >
> > Thank the reviewer for this comment.
> >
> > Firstly, the problem settings of this paper and the reference paper [2] suggested by the reviewer are different. As we stated in Section 2, the problem we considered is how to maximize the total value from the perspective of a single advertiser, which is naturally a single-agent problem. What we argue in Fig. 2(b) is that all other advertisers acting as a part of the environment are not correctly modeled in the VAS, which will cause the IBOO. In addition, our method can solve the IBOO, including the inaccurate market price changing issue.
> > However, the problem in [2] studies how to realize multi-objective goals, involving the interests of advertisers and the platform, which is a multi-agent problem. Hence, it is not very proper to compare our algorithm with the method in [2] that solves a different problem.
> >
> > Secondly, in the original manuscript, we have compared our algorithm with the USCB [1], the SOTA method of the single-agent auto-bidding problem. We also note that USCB was proposed in SIGKDD 2021, the same as the reference [2]. Besides, both USCB and the method in [2] suffer from the IBOO.
> >
> > Nonetheless, we conduct the A/B test between our algorithm and the method in [2] in the real-world advertising system, and the results are shown below.
> >
> >        Algorithms      |   ROI      |     BuyCnt   |   CPA    |     ConBgd
> >        Method in [2]   |   2.79     |    121,616   |   44.86  |    5,455,507.35
> >        Our method      |   3.15     |    139,599   |  40.05   |    5,590,858.50
> >                        |   +12.90%  |    +14.79%   | -10.72%  |     +2.48%
> >
> > We can see that our method outperforms the method in [2] in all four metrics. We have cited the reference paper [2] in the revised manuscript.
> >
> > [1] "A Unified Solution to Constrained Bidding in Online Display Advertising." SIGKDD, 2021.
> >
> > [2]"Multi-Agent Cooperative Bidding Games for Multi-Objective Optimization in e-Commercial Sponsored Search." SIGKDD, 2021.
> >
> > **Minor issues**
> >
> > Thank the reviewer for this suggestion. We have polished the presentation of the paper and improved the writing. The modified parts are highlighted in blue in the revised manuscript.
> >
> > To the best of our knowledge, $\mu$ and $\pi$ are often used in RL literature to represent deterministic and stochastic policies, respectively. In this paper, we use $\mu$ to denote the target deterministic policy and use $\pi$ to denote the stochastic exploration policy.
> >
> > In addition. we have modified "derivations"->"deviations", "are in four-fold"->"is fourfold", "is substitute by" -> "is substituted by".
> >
> > **Questions**
> >
> > Thank the reviewer for the comments on the details of the paper. Our answers are listed below.
> >
> > **(1) What do you mean by $p_{j,t}$ and $v_{j,t}$ in proposition 2? Do you mean $p^2_{j,t}$ and $v^2_{j,t}$?**
> >
> > Note that $p_{j,t}$ and $v_{j,t}$ are both defined in Section 2 in the original manuscript, meaning the real market price and value of impression $j$, respectively. $p_{j,t}^2$ and $v_{j,t}^2$ are estimated market price and value of impression $j$ at stage 2.
> >
> > **(2)What do you mean by $\mu_s$ in Figure 6?**
> >
> > $\mu_s$ is the initial safe policy for warm booting. It has been mentioned for the first time in Section 3 in the original manuscript.
> >
> > **(3) Does $\pi_e$ policy mean the SER policy after 7 iterations?**
> >
> > $\pi_e$ is derived as the general form of the SER policy. As the proposed method works in an iterative manner, we design the SER policy $\pi_{e,\tau}$ at the $\tau-th$ iteration based on $\pi_e$. $\pi_e$ equals to the SER policy at the 1st iteration $\pi_{e,1}$, but does not equal the SER policies in the subsequent iterations since they have been modified.
> >
> > **(4) Why do you only show results after three iterations in Table 3?**
> >
> > Note that we cannot know the optimal policy in a real-world advertising system. Hence, we cannot judge if the target policy achieves the optimal policy in real-world experiments. Nonetheless, we can know the optimal policy in the simulated experiments, and as shown in Fig. 8(b) in the original manuscript, our method achieves the optimal policy at the 5-th iteration. This shows the effectiveness of our method. Moreover, we have conducted two more iterations of our method compared with the USCB in real-world experiments, and the results are shown below. We can see that the target policy continues to improve, which further validates the effectiveness of our method.
> >
> >          | iterations  |      ROI     |     BuyCnt    |    CPA      |     ConBgd
> >          |  4-th       |     1.08%    |     3.02%     |    -1.79%   |     1.43%
> >          |  5-th       |     0.83%    |     2.79%     |    -1.54    |     1.32%

---

> > > ### Author Response · Authors · 2022-08-09
> > > **Response to Reviewer o6ML**
> > >
> > > We kindly ask the reviewer to reassess the paper in light of these comments if they clear things up. We are happy to answer more if you have any remaining concerns or questions.

---

> > > > ### Comment · Reviewer_o6ML · 2022-08-09
> > > > **Sorry for the delay**
> > > >
> > > > I have carefully read and verified your revision and reponses. The major concerns are well addressed. I am going to raise the score.

---

> > > > > ### Author Response · Authors · 2022-08-10
> > > > > **Thank you very much for the reply**
> > > > >
> > > > > Thank you very much for your reply and the acknowledgement of our responses! We really appreciate your value feedbacks to our work! Thank you again, and looking forward to your final rating.

---

> > > > > > ### Author Response · Authors · 2022-08-10
> > > > > > **Thank you very much**
> > > > > >
> > > > > > Thank you very much for your reply and the final rating. We really appreciate the time and effort that you have dedicated to our paper!

---

### Official Review · Reviewer_7gxf · 2022-07-11

**Rating:** 6
**Confidence:** 3
**Soundness:** 3 good
**Presentation:** 2 fair
**Contribution:** 3 good

**Summary:**

The paper tries to address the problem of inconsistency between the online and offline settings in the outbidding problem with safeguards. It proposes an online RL framework to improve the auto-bidding policies through direct interactions with the real environment. An online exploration policy is discussed trying to address safety and efficiency requirements.


**Questions:**

- It seems the discussion of the related literature is not very clear. For example, it is stated that if one trains the policy using data collected by a certain behavior policy directly from the RAS, then there will be certain extrapolation errors that degrade the policy performance. More discussion should be given here to avoid confusion. It is also helpful to be more specific and have some discussion on how this paper differs from the other safe online RL methods that try to explore the environment. It is also stated that there are methods using constraint functions, how does these methods related to this problem, and specifically why are they not good candidates?

- In the algorithm section, the algorithm details need more elaboration, also the description of the warm boosting section can be made more clear.


**Ethics Review Area:**

["I don’t know"]

**Limitations:**

Not applicable.

**Strengths And Weaknesses:**

Strength: The paper tries to address the problem of inconsistency between the online and offline settings in the outbidding problem with safeguards, and the experiments look encouraging.

Weakness: In certain places, the presentation can be made clearer, and the algorithm details need more elaboration.

---

> ### Author Response · Authors · 2022-08-02
> **Response to Reviewer 7gxf**
>
> Thank you for your detailed review and valuable feedback to our work. Below we answer the comments and suggestions given in this review.
>
> **1. Weakness: In certain places, the presentation can be made clearer, and the algorithm details need more elaboration.**
>
> Thank the reviewer for this suggestion. We have polished the presentations and elaborated our algorithms in more details in the revised manuscript.
>
> **2. It seems the discussion of the related literature is not very clear.**
>
> Thank the reviewer for this concern. We have added more detailed discussion of the related literature in Section 1 and Appendix B in the revised manuscript. The main contents are summarized as follows.
>
> **(2.1) More discussion on certain extrapolation errors:** Extrapolation error is the main challenge the standard RL methods face when training with a fixed dataset. Specifically, if we train the policy with standard RL methods (e.g. DDPG [1]) based on a fixed dataset collected by some behavior policies, there will be extrapolation error that can seriously degrade the policy performance. Formally, extrapolation error means the method's misestimation of the states and actions outside the fixed dataset. A typical misestimation happens to the Q function. As the fixed dataset cannot contain all the data (since its amount can usually be infinite) from the environment, the trained Q function can only be accurate at the states and actions inside the dataset, while be inaccurate (usually overestimated) at those outside the dataset. This will make the actor network learn actions that extremely deviate from the behavioral actions and often bias towards bad actions. Hence, the policy performance can be seriously degraded. Offline RL algorithms [2,3,4] targets to address the extrapolation error challenge through conservative updating approaches, including policy constraints and conservative penalties, which are represented in Appendix B in the revised manuscript.
>
> **(2.2) How this paper differs from the other safe online RL methods that try to explore the environment:**  The method in this paper differs from other works in two aspects. Firstly, the definition of safety in our method differs from that in many existing works [5,6,7,8]. In those works, there exists a specific set of safe (dangerous) state-action pairs that can (cannot) be explored. Some work [5] even requires to know the safe states in advance. However, in the auto-bidding problem, no specific actions at any state cannot be explored as long as the expected accumulative rewards maintain at a high level. This requires the exploration policy to always keep high performance during the whole training process, which is more challenging. Secondly, the design of our exploration policy differs from other works. For example, [6] realizes safe explorations by adding a safety layer at the end of the actor network. However, the safety layer needs to be trained by a prior dataset and can be inaccurate at states outside the dataset. [7] uses offline safety tests to examine the safety of the latest policy and directly applies it to explore online if it passes the tests. However, as we stated in Appendix G.2, there is no such reliable offline safety test in auto-bidding. Besides, [8] realizes safe explorations by gradually increasing the attraction regions of the initial safe policy. However, it leverages the assumption that the environment is a linear model, which is not suitable for auto-bidding. As for this paper, based on the proved Lipschitz property of the Q function, we design the exploration policy by offsetting the actions to the promising directions relative to an initial safe policy. The proposed method is typically suitable for the auto-bidding problem we considered.
>
> **(2.3) How does methods using constraint functions related to this problem, and specifically why are they not good candidates:** There are mainly two kinds of methods using constraint functions, including the reward shaping (or Lagrange multiplier) method [9] and methods based on CMDP [10]. The first kind of methods are not good candidates since it can only guarantee the safety during test phase. However, we need to design the exploration policy that is required to be safe during training phase. The second kind of methods are developed based on the CMDP, where there is a constraint (or safety) function $C$ for each state, action or state-action pair. For example, in [10], the constraint function is $C: \mathcal{S}\rightarrow \{ 0,1 \}$. However, as we stated in Section 3 in the revised manuscript, it is not proper to design $C$ as only a function of states or actions. Hence, the second kinds of methods are also not good candidates.

---

> > ### Author Response · Authors · 2022-08-02
> > **Response to Reviewer 7gxf**
> >
> > **3. In the algorithm section, the algorithm details need more elaboration, also the description of the warm boosting section can be made more clear.**
> >
> > Thank you for this suggestion. We have added more details of the proposed algorithm in the revised manuscript, as well as the theoretical proof of the algorithm. Here we summarize the proposed algorithm and the warm booting part. The SORL works in an iterative manner. In each iteration $\tau$, the SORL collects data directly from the RAS with the proposed exploration policy $\pi_{e,\tau}$ and uses the V-CQL method to train the target policy $\mu_\tau$ with the newly collected data. Specially, in the first (i.e., $0$-th) iteration, we need a known policy to start the data collection process, and thus, boot the SORL. As the policy will directly interact with the RAS, it should be safe. Hence, we leverage the state-of-the-art auto-bidding policies, for example, USCB that has already been deployed to many RAS in practice, to start data collections in the first iteration. This is the warm booting. Besides, as demonstrated in the manuscript, the subsequent exploration policies $\pi_{e,\tau}$ is guaranteed to be safe. Moreover, due to the constantly feedback of the collected data, the target policy $\mu_\tau$ will be improved.
> >
> > **4. Limitations: Not applicable**
> >
> > Thank the reviewer for the concerns of the applicability of our algorithm. As stated in Section 5 in the original manuscript, we have applied our algorithm to a real-world advertising system on Taobao, one of the world’s largest E-commerce platforms. The experiment results show that the proposed algorithm outperforms the state-of-the-art auto-bidding methods.
> >
> > [1] "Continuous control with deep reinforcement learning", 2016.
> >
> > [2] "Off-policy deep reinforcement learning without exploration", 2019.
> >
> > [3] "Conservative q-learning for offline reinforcement learning", 2020.
> >
> > [4] "Stabilizing off-policy q-learning via bootstrapping error reduction", 2019.
> >
> > [5] "Safe exploration in finite markov decision processes with gaussian Processes", 2018.
> >
> > [6] "Safe exploration in continuous action space", 2018.
> >
> > [7] "Safe policy improvement with baseline bootstrapping", 2019.
> >
> > [8] "Safe model-based reinforcement learning with stability guarantees", 2017.
> >
> > [9] "Batch policy learning under constraints", 2019.
> >
> > [10] "Conservative safety critics for exploration", 2021.

---

> > > ### Comment · Reviewer_7gxf · 2022-08-07
> > > **Thanks for the response**
> > >
> > > Thanks for the detailed response and clarification! I will raise the score accordingly.

---

> > > > ### Author Response · Authors · 2022-08-10
> > > > **Thank you very much for the reply**
> > > >
> > > > Thank you very much for the reply! We really appreciate the suggestions in the review! They are very useful and help us make the paper stronger.

---

### Official Review · Reviewer_qefY · 2022-07-14

**Rating:** 7
**Confidence:** 3
**Soundness:** 3 good
**Presentation:** 3 good
**Contribution:** 3 good

**Summary:**

In this paper, the authors provide a new online bidding algorithm using reinforcement learning and a safety function to accommodate the mismatch between online and offline data (IBOO challenge in the paper). The authors characterize the lipshitzness of the safety function and validate the effectiveness of the proposed algorithm through synthetic data and real advertising data.

**Questions:**

If I understand correctly, one of the main causes to IBOO problem is the mismatch of the data resource for RAS and VAS. Could the authors provide more details why we can only collect data in stage 2 in VAS?

**Strengths And Weaknesses:**

Strengths: This paper investigates a very important problem in practice. Auto-bidding is getting more and more popular in real production system and this paper provides a very reasonable approach to design real time bidding algorithm. The paper is well-written and clearly states the contribution. In addition, the authors show their algorithm can achieve very good performance empirically. I have several question as follows.

Weaknesses: There is no theoretical guarantee for the proposed bidding algorithm, however, this doesn't really hurt the contribution of this paper since the authors show that their algorithm achieves good performance over real advertising.

---

> ### Author Response · Authors · 2022-08-02
> **Response to Reviewer qefY**
>
> We appreciate the time and effort that the reviewer has dedicated to providing valuable feedback on our manuscript. We next address the concerns raised in the review.
>
> **1. There is no theoretical guarantee for the proposed bidding algorithm.**
>
> Thank you for the valuable feedback. In the original manuscript, we have theoretically proved the safety of the designed safety zone, based on which we develop the exploration policy. To make the theory more general, we have added the theoretical guarantee of safety directly for the proposed online exploration policy in Theorem 3 in the revised manuscript, and presented the detailed proofs in Appendix E.3 in the revised manuscript. Specifically, denote the
> total accumulated value of the advertiser with the proposed exploration policy $\pi_{e}$ (can be $\pi_{e,\mathcal{N}}$, $\pi_e$ or $\pi_{e,\tau}$) and with safe policy $\mu_s$ as $V(\pi_e)$ and $V(\mu_s)$, respectively.
> We prove that the difference between $V(\pi_e)$ and $V(\mu_s)$ is not bigger than a certain upper bound $\xi\gamma^{t_1}L_Q\Delta T$. As the safety zone range satisfies $\xi\le\frac{\epsilon_s}{L_Q}$, we can guarantee that
> \begin{align}
> |V(\pi) - V(\mu_s)|\le\epsilon_s\gamma^{t_1}\Delta T.
> \end{align}
> Hence, we have theoretically guarantee the safety of the proposed online exploration policy.
> In addition, we have conducted both simulated and real-world experiments in Section 5 and Appendix G.3.3 to validate the safety of the proposed exploration policy and the effectiveness of the proposed V-CQL method. The results show that our methods largely outperform the state-of-the-art bidding method.
>
> **2. Could the authors provide more details why we can only collect data in stage 2 in VAS.**
>
> Thank you for the suggestion. As the reviewer correctly points out, one of the main causes of the IBOO problem is the mismatch of the data resource for the RAS and VAS. Specifically, the VAS is usually built only based on the data generated in stage 2 of the RAS in practice. This is because the amount of data generated in stage 1 of the RAS is very large (about $10^2$ to $10^4$ impression opportunities coming to stage 1 at every moment, and about $10^6$ advertisers bidding for each impression opportunity), and it is computationally infeasible to build the VAS based on data in stage 1. We have added this explanation to Appendix A.1.2 in the revised manuscript.

---

### Meta-Review · Area_Chair_iLuh · 2022-08-29

**Recommendation:** Accept
**Confidence:** Certain

**Metareview:**

The authors address the issue of inconsistencies between modeled and training data for auto bidding RL policies and demonstrate the efficacy of their approach both analytically and experimentally.

**Award:**

No

---

### Decision · Program_Chairs · 2022-09-14

Accept